# XLand-MiniGrid: Scalable Meta-Reinforcement Learning Environments in JAX

**Alexander Nikulin**[†]
AIRI, MIPT
nikulin@airi.net

**Vladislav Kurenkov**[†]
AIRI, Innopolis University
kurenkov@airi.net

**Ilya Zisman**[†]
AIRI, Skoltech
zisman@airi.net

**Artem Agarkov**[†]
MIPT
agarkov.as@phystech.edu

**Viacheslav Sinii**
T-Bank
v.siniy@tbank.ru

**Sergey Kolesnikov**
T-Bank
s.s.kolesnikov@tbank.ru

## Abstract

Inspired by the diversity and depth of XLand and the simplicity and minimalism of MiniGrid, we present XLand-MiniGrid, a suite of tools and grid-world environments for meta-reinforcement learning research. Written in JAX, XLand-MiniGrid is designed to be highly scalable and can potentially run on GPU or TPU accelerators, democratizing large-scale experimentation with limited resources. Along with the environments, XLand-MiniGrid provides pre-sampled benchmarks with millions of unique tasks of varying difficulty and easy-to-use baselines that allow users to quickly start training adaptive agents. In addition, we have conducted a preliminary analysis of scaling and generalization, showing that our baselines are capable of reaching millions of steps per second during training and validating that the proposed benchmarks are challenging. XLand-MiniGrid is open-source and available at https://github.com/corl-team/xland-minigrid.

## 1 Introduction

Reinforcement learning (RL) is known to be extremely sample inefficient and prone to overfitting, sometimes failing to generalize to even subtle variations in environmental dynamics or goals (Rajeswaran et al., 2017; Zhang et al., 2018; Henderson et al., 2018; Alver & Precup, 2020). One way to address these shortcomings are meta-RL approaches, where adaptive agents are pre-trained on diverse task distributions to significantly increase sample efficiency on new problems (Wang et al., 2016; Duan et al., 2016). With sufficient scaling and task diversity, these approaches are capable of astonishing results, reducing the adaptation time on new problems to human levels and beyond (Team et al., 2021, 2023).

At the same time, meta-RL methods have major limitations. Since the agent requires thousands of different tasks for generalization, faster adaptation during inference comes at the expense of significantly increased pre-training requirements. For example, a single training of the Ada agent (Team et al., 2023) takes five weeks[3], which can be out of reach for most academic labs and practitioners. Even those who might have the training resources would still be unable to use them, as the XLand environment is not publicly available. We believe, and this also has been pointed out by Wang et al. (2021), that such demanding requirements are the reason why most recent works on adaptive agents (Laskin et al., 2022; Lee et al., 2023; Lu et al., 2023; Norman & Clune, 2023) avoid complex environments in favor of more simplistic ones (e.g., simple navigation).

---

[†]Work done while at T-Bank
[3]According to Appendix D.2 in Team et al. (2023).

38th Conference on Neural Information Processing Systems (NeurIPS 2024) Track on Datasets and Benchmarks.

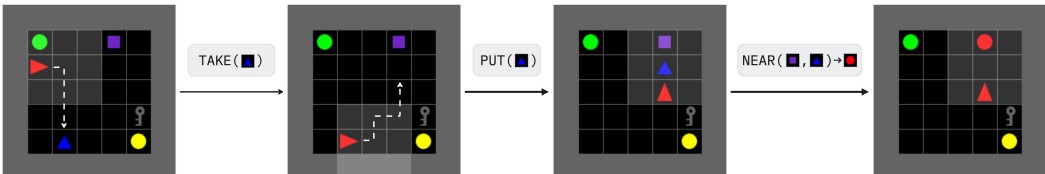

Figure 1: Visualization of how the production rules in XLand-MiniGrid work, exemplified by a few steps in the environment. In the first steps, the agent picks up the blue pyramid and places it next to the purple square. The NEAR production rule is then triggered, which transforms both objects into a red circle. See Figure 2 and Section 2.1 for additional details.

While simple environments are an affordable and convenient option for theoretical analysis, they are not enough for researchers to discover the limits and scaling properties of proposed algorithms in practice. To make such research more accessible, we continue the successful efforts of Freeman et al. (2021); Lu et al. (2022); Bonnet et al. (2023); Koyamada et al. (2023) at accelerating environments using JAX (Bradbury et al., 2018), and introduce the XLand-MiniGrid, a library of grid world environments and benchmarks for meta-RL research (Section 2). We carefully analyze the scaling properties (Section 4.1) of environments and conduct preliminary experiments to study the performance and generalization of the implemented baselines, validating that the proposed benchmarks are challenging (Section 4.2).

## 2   XLand-MiniGrid

In this paper, we present the initial release of **XLand-MiniGrid**, a suite of tools, benchmarks and grid world environments for meta-RL research. We do not compromise on task complexity in favor of affordability, focusing on democratizing large-scale experimentation with limited resources. XLand-MiniGrid is open-source and available at `https://github.com/corl-team/xland-minigrid` under Apache 2.0 license.

Similar to XLand (Team et al., 2023), we introduce a system of extensible rules and goals that can be combined in arbitrary ways to produce diverse distributions of tasks (see Figures 1 and 2 for a demonstration). Similar to Mini-Grid (Chevalier-Boisvert et al., 2023), we focus on goal-oriented grid world environments and use a visual theme already well-known in the community. However, despite the similarity, XLand-MiniGrid is written from scratch in the JAX framework and can therefore run directly on GPU or TPU accelerators, reaching millions of steps per second with a simple `jax.vmap` transformation. This makes it possible to use the Anakin architecture (Hessel et al., 2021) and easily scale to multiple devices using the `jax.pmap` transformation.

In addition to environments, we provide pre-sampled benchmarks with millions of unique tasks, simple baselines with recurrent PPO (Schulman et al., 2017) that scale to multi-GPU

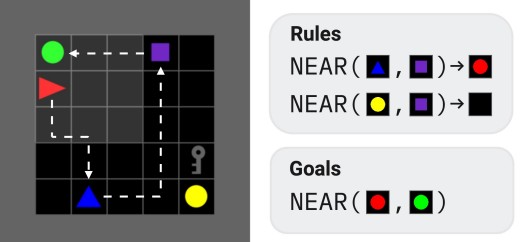

Figure 2: Visualization of a specific sampled task (see Figure 3) in XLand-MiniGrid. We highlighted the optimal path to solve this particular task. The agent needs to take the blue pyramid and put it near the purple square in order to transform both objects into a red circle. To complete the goal, a red circle needs to be placed near the green circle. However, placing the purple square near the yellow circle will make the task unsolvable in this trial. Initial positions of objects are randomized on each reset. Rules and goals are hidden from the agent.

setups, walk-through guides that explain the API, and colab notebooks that make it easy for users to start training adaptive agents. We hope that all of this will help researchers quickly start experimenting.

This section provides a high-level overview of the library, describing the system of rules and goals, the observation and action spaces, and its API. The implemented environments are also described.

## 2.1 Rules and Goals

In XLand-MiniGrid, the system of rules and goals is the cornerstone of the emergent complexity and diversity. In the original MiniGrid (Chevalier-Boisvert et al., 2023), some environments have dynamic goals, but the dynamics themselves are never changed. To train and evaluate highly adaptive agents, we need to be able to change the dynamics in non-trivial ways (Team et al., 2023).

**Rules.** Rules are functions that can change the environment state in a deterministic fashion according to the given conditions. For example, the `NEAR` rule (see Figure 1 for a visualization) accepts two objects `a`, `b` and transforms them to a new object `c` if `a` and `b` end up on neighboring tiles. Rules can change between resets. For efficiency reasons, the rules are evaluated only after some actions or events occur (e.g., the `NEAR` rule is checked only after the `put_down` action).

**Goals.** Goals are similar to rules, except they do not change the state, only test conditions. For example, the `NEAR` goal (see Figure 1 for a visualization) accepts two objects `a`, `b` and checks that they are on neighboring tiles. Similar to rules, goals are evaluated only after certain actions and can change between resets.

Both rules and goals are implemented with classes. However, in order to be able to change between resets and still be compatible with JAX, both rules and goals are represented with an array encoding, where the first index states the rule or goal `ID` and the rest are arguments with optional padding to the same global length. Thus, every rule and goal should implement the `encode` and `decode` methods. The environment state contains only these encodings, not the actual functions or classes. For the full list of supported rules and goals, see Appendix I.

## 2.2 API

**Environment interface.** There have been many new JAX-based environments appearing recently (Freeman et al., 2021; Lange, 2022; Bonnet et al., 2023; Koyamada et al., 2023; Rutherford et al., 2023; Jiang et al., 2023; Frey et al., 2023; Lechner et al., 2023), each offering their own API variations. The design choices in most of them were not convenient for meta-learning[4], hence why we decided to focus on creating a minimal interface without making it general. The core of our library is interface-independent, so we can quickly switch if a unified interface becomes available in the future. If necessary, practitioners can easily write their own converters to the format they need, as has been done in several projects[5] that already use XLand-MiniGrid.

At a high level, the current API combines `dm_env` (Muldal et al., 2019) and `gymnax` (Lange, 2022). Each environment inherits from the base `Environment` and implements the jit-compatible `reset` and `step` methods, with custom `EnvParams` if needed. The environment itself is completely stateless, and all the necessary information is contained in the `TimeStep` and `EnvParams` data classes. This design makes it possible for us to vectorize on arbitrary arguments (e.g., rulesets) if their logic is compatible with jit-compilation. Similar to `Gym` (Brockman et al., 2016), users can register new environment variants with custom parameters to conveniently reuse later with the help of `make`. We provide minimal sample code to instantiate an environment from the registry, reset, step and optionally render the state (see Listing 1). For an example of how to compile the entire episode rollout, see Appendix D.

**State and TimeStep.** Similar to `dm_env`, `TimeStep` contains all the information available to the agent, such as `observation`, `reward`, `step_type` and `discount`. The step type will be FIRST at the beginning of the episode, LAST at the last step, and MID for all others. The discount can be in the $[0, 1]$ range, and we set it to 0.0 to indicate the end of the episode or trial. In addition, it contains `State` with all the necessary information to describe the environment dynamics, such as the grid, agent states, encoding of rules and goals, and a key for the random number generator that can be used during resets. The combination of the internal state with the timestep is a bit different from the previous designs by Lange (2022); Bonnet et al. (2023), but it allows for some simplifications, such as an auto-reset wrapper implementation based on the current or previous step (in the style of Gym (Brockman et al., 2016) or EnvPool (Weng et al., 2022)).

---

[4]For example the most popular Jumanji library currently does not allow changing env parameters during reset, see `https://github.com/instadeepai/jumanji/issues/212`.

[5]See Stoix, varibad_jax, MetaLearnCuriosity or ARLBench codebases.

```
import jax
import xminigrid

reset_key = jax.random.key(0)
# to list available environments:
xminigrid.registered_environments()

# create env instance
env, env_params = xminigrid.make("XLand-MiniGrid-R9-25x25")
# change some default params
env_params = env_params.replace(max_steps=100)

# fully jit-compatible step and reset methods
timestep = jax.jit(env.reset)(env_params, reset_key)
timestep = jax.jit(env.step)(env_params, timestep, action=0)

# optionally render the state
env.render(env_params, timestep)
```

Listing 1: Basic example usage of XLand-MiniGrid.

**Observation and action space.** Although not fully compatible, we made an effort to be consistent with the original MiniGrid. Observations describe a partial field of view around the agent as two-dimensional arrays, where each position is encoded by the tile and color IDs. Thus, observations are not images and should not be treated as such by default. While naively treating them as images can work in most cases, the correct approach would be to pre-process them via embeddings. If necessary, practitioners can render such observations as images via the wrapper, although with some performance overhead (see Appendix H). We also support the ability to prohibit an agent from seeing through walls. The agent actions, namely `move_forward`, `turn_left`, `turn_right`, `pick_up`, `put_down`, `toggle`, are discrete. The agent can only pick up one item at a time, and only if its pocket is empty. In contrast to Team et al. (2023), the actual rules and goals of the environment remain hidden from the agent in our experiments. However, our environment does not impose any restrictions on this, and practitioners can easily access them if needed (see our preliminary experiments in such a setting in Appendix G).

### 2.3 Supported Environments

In the initial release, we provide environments from two domains: XLand and MiniGrid. XLand is our main focus, and for this domain, we implement single-environment `XLand-MiniGrid` and numerous registered variants with different grid layouts and sizes (see Figure 14 in the Appendix). All of them can be combined with arbitrary rulesets from the available benchmarks (see Section 3) or custom ones, and follow the naming convention of `XLand-MiniGrid-R{#rooms}-{size}`. We made an effort to balance the limit on the maximum number of steps so that the tasks cannot be brute-forced by the agent on every trial without using any memory. Thus, we use $3$ x grid height x grid width as a heuristic to set the default maximum number of steps, but this can be changed afterwards if needed. While meta-RL is our main motivation, this environment can be useful for research in exploration, continual learning, unsupervised RL or curriculum learning. For example, we can easily model novelty as a change in rules, goals or objects between episodes, similar to NovGrid (Balloch et al., 2022).

Furthermore, due to the generality of the rules and goals and the ease of extensibility, most non-language-based tasks from the original MiniGrid can also be quickly implemented in XLand-MiniGrid. To demonstrate this, we have ported the majority of such tasks, including the popular `Empty`, `FourRooms`, `UnlockPickUp`, `DoorKey`, `Memory` and others. For a full list of registered environments, see Appendix L (38 in total).

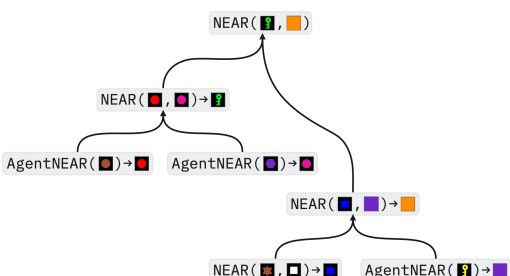

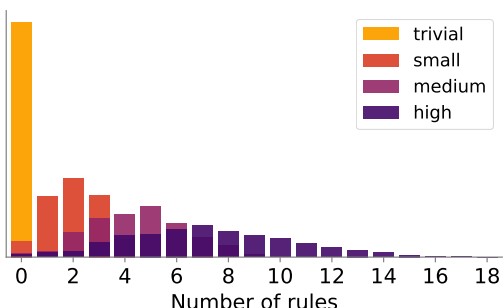

Figure 3: Visualization of a specific task tree with depth two, sampled according to the procedure described in Section 3. The root of the tree is a goal to be achieved by the agent, while all other nodes are production rules describing possible transformations. At the beginning of each episode, only the input objects of the leaf production rules are placed on the grid. In addition to the main task tree, the distractor production rules can be sampled. They contain already used objects to introduce dead ends. All of this together is what we call a **ruleset**, as it defines the task.

Figure 4: Distribution of the number of rules for the available benchmark configurations. One can see that each successive benchmark offers an increasingly diverse distribution of tasks, while still including tasks from the previous benchmarks. The average task complexity, as well as tree depth, also increases. See Section 3 for the generation procedure and Appendix J for the exact generation configuration. Besides, users can generate and load custom benchmarks easily, even with a custom generation procedure, as long as the final format is the same.

## 3 Benchmarks

Although users can manually compose and provide specific rulesets for environments, this can quickly become cumbersome. Therefore, for large-scale open-ended experiments, we provide a way to procedurally generate tasks of varying diversity and complexity, which can in turn generate thousands of unique rulesets. However, since the generation procedure can be quite complex, representing it in a form that is convenient for efficient execution in JAX is not feasible. To avoid unnecessary overhead and standardize comparisons between algorithms and experiments, we pre-generated several benchmarks with up to three million unique rulesets. The generation process itself as well as available configurations are described in detail below.

**Generation Procedure.** For the generation procedure, we closely follow the approach of Team et al. (2023). A similar procedure was also used in a concurrent work by Bornemann et al. (2023) but for multi-agent meta-learning. On a high level, each task can be described as a tree (see Figure 3), where each node is a production rule and the root is the main goal. Generation starts by uniformly sampling an agent's goal from the ones available. Then, new production rules are sampled recursively at each level so that their output objects are the input objects of the previous level. Since all rules have at most two arguments, the tree will be a complete binary tree in the worst-case scenario. At the start of the episode, only objects from the leaf rules are placed on the grid, and their positions are randomized at each reset. Thus, to solve the task, the agent has to trigger these rules in a sequence to get the objects needed for the goal. This hierarchical structure is very similar to the way tasks are organized in the famous Minecraft (Guss et al., 2021) or the simpler Crafter (Hafner, 2021) benchmarks. For object sampling, we used ten colors and seven tile types (e.g., circle, square).

When sampling production rules, we restrict the possible choices of input objects, excluding those that have already been used. This is done so that the objects are present only once as input and once as output in the main task tree. To increase diversity, we also added the ability to sample depth instead of using a fixed one, as well as branch pruning. With some probability, the current node can be marked as a leaf and its input objects added as initial objects. This way, while using the same budget, we can generate tasks with many branches, or more sequential ones but with greater depth.

To prevent the agent from indiscriminately triggering all production rules and brute forcing, we additionally sample distractor rules and objects. Distractor objects are sampled from the remaining objects and are not used in any rules. Distractor production rules are sampled so that they use objects from the main task tree, but never produce useful objects. This creates dead ends and puts the game

in an unsolvable state, as all objects are only present once. As a result, the agent needs to experiment intelligently, remembering and avoiding these rules when encountered.

**Available configurations.** In the initial release, we provide four benchmark types with various levels of diversity: `trivial`, `small`, `medium` and `high`. For each benchmark type, one million unique rulesets were pre-sampled, including additional variations with three millions for `medium` and `high`. Generally, we tried gradually increasing diversity to make it so that each successive benchmark also includes tasks from the previous ones (see Figure 4). Thus, due to the increasing task-tree depth, the average difficulty also increases. For example, the `trivial` benchmark has a depth of zero and can be used for quick iterations and debugging. For exact generation settings, see Table 4 in Appendix J.

For ease of use when working with benchmarks, we provide a user-friendly interface that allows them to be downloaded, sampled, split into train and test sets (see an extended usage example in Appendix D). Similar to Fu et al. (2020); Kurenkov et al. (2023), our benchmarks are hosted in the cloud and will be downloaded and cached the first time they are used. In addition, we also provide a script used for generation, with which users can generate and load their own benchmarks with custom settings.

# 4 Experiments

In this section, we demonstrate XLand-MiniGrid's ability to scale to thousands of parallel environments, dozens of rules, various grid sizes, and multiple accelerators (see Section 4.1). In addition, we describe the implemented baselines and validate their scalability. Finally, we perform preliminary experiments showing that the proposed benchmarks, even with minimal diversity, present a significant challenge for the implemented baseline, leaving much room for improvement, especially in terms of generalization to new problems (see Section 4.2). For each experiment, we list the exact hyperparameters and additional details in Appendix K.

## 4.1 Scalability

**Simulation throughput.** Figure 5a shows the simulation throughput for a random policy averaged over all registered environment variations (38 in total, see Section 2.3). All measurements were done on A100 GPUs, taking the minimum value among multiple repeats. In contrast to Jumanji (Bonnet et al., 2023), we had an auto-reset wrapper enabled to provide estimates closer to real use, as resetting can be expensive for some environments. For meta-RL environments, random rulesets from the `trivial-1m` benchmark were sampled. One can see that the scaling is almost log-log linear with the number of parallel environments, although it does begin to saturate around $2^{13}$ on a single device. However, on multiple devices, scaling remains log-log linear without signs of saturation, easily reaching tens of millions of steps per second. We provide the scripts we used so that our results can be replicated on other hardware. As an example, we replicated some of the benchmarks on consumer grade GPUs like 4090 in the Appendix F.

**Scaling grid size.** While most of MiniGrid's grid world environments (Chevalier-Boisvert et al., 2023) use small grid sizes, it is still interesting to test the scaling properties of XLand-MiniGrid in this dimension, as larger sizes may be needed for difficult benchmarks to fit all the initial objects. As one can see in Figure 5b, the simulation throughput can degrade significantly with increasing grid size, and can also show earlier signs of saturation. A possible explanation for this phenomenon is that many game loop operations, such as conditional branching during action selection, do not fit well with the parallelism principles of the JAX framework. Similar results have been observed in previous works using JAX-based environments (Bonnet et al., 2023; Koyamada et al., 2023). Nevertheless, the throughput remains competitive even at larger sizes. Furthermore, as shown in Figure 5d, the throughput can be considerably improved with multiple devices, allowing a larger pool of parallel environments and mitigating saturation. When it comes to small grid sizes[6], they can still be a significant challenge for existing algorithms (Zhang et al., 2020), which we will also demonstrate in Section 4.2.

**Scaling number of rules.** According to Team et al. (2023), a full-scale XLand environment can use more than five rules. Similarly, our benchmarks can contain up to eighteen rules (see Figure 4). To test XLand-MiniGrid under similar conditions, we report the simulation throughput with different

---

[6]Note that, due to how the rules and goals are encoded, the maximum grid size is currently limited to 255.

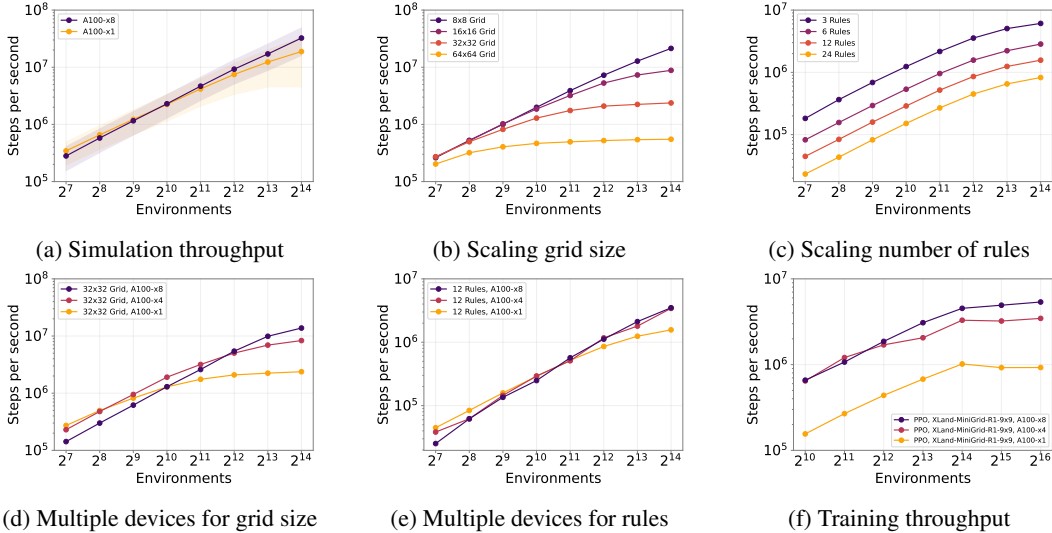

| | | |
|---|---|---|
| (a) Simulation throughput | (b) Scaling grid size | (c) Scaling number of rules |
| (d) Multiple devices for grid size | (e) Multiple devices for rules | (f) Training throughput |

Figure 5: Analysis of XLand-MiniGrid's ability to scale to thousands of parallel environments, dozens of rules, different grid sizes and multiple accelerators. All measurements were performed on A100 GPUs, taking the minimum between multiple attempts with the auto-reset wrapper enabled and with random policy, except for **(d)**. **(a)** Simulation throughput averaged over all registered environment variations (38 in total, see Section 2.3). Scaling is almost log-log linear, with slight saturation on a single device. **(b)** Increasing the grid size can significantly degrade simulation throughput, leading to earlier saturation. **(c)** Increasing the number of rules monotonically reduces the simulation throughput, with no apparent saturation. To maintain the same level of throughput, the number of parallel environments should be increased. **(d)**-**(e)** Parallelization across multiple devices can mitigate saturation and significantly increase throughput, even at large grid sizes and rule counts. **(f)** Training throughput for the implemented recurrent PPO baseline (see Section 4.2). The PPO hyperparameters, except for model size and RNN sequence length, were tuned for each setup to maximize utilization per device. Single device training throughput saturates near one million steps per second. Similarly to random policy, it can be increased greatly with multiple devices. We additionally provide figures without log-axes at Appendix E.

numbers of rules (see Figure 5c). For testing purposes, we simply replicated the same NEAR rule multiple times and used a grid size of 16x16. As one can see, the throughput decreases monotonically as the number of rules increases. As a result, the number of parallel environments must be increased to maintain the same throughput level. However, in contrast to increasing the grid size, there is no apparent saturation even at 24 rules. The throughput can be improved even further by increasing the number of devices (see Figure 5e).

## 4.2 Baselines

With the release of XLand-MiniGrid, we are providing near-single-file implementations of recurrent PPO (Schulman et al., 2017) for single-task environments and its extension to $RL^2$ (Duan et al., 2016; Wang et al., 2016) for meta-learning as baselines. The implementations were inspired by the popular PureJaxRL (Lu et al., 2022), but extended to meta-learning and multi-device setups. Due to the full environment compatibility with JAX, we can use the Anakin architecture (Hessel et al., 2021), jit-compile the entire training loop, and easily parallelize across multiple devices using `jax.pmap` transformation. We also provide standalone implementations in the colab notebooks.

While our implementation is greatly simplified in comparison to Ada (Team et al., 2023), it encapsulates the main principles and is easy to understand and modify, e.g., swapping simple GRU (Cho et al., 2014) for more modern and efficient state space models (Lu et al., 2023). Next, we perform preliminary experiments to test the scaling, performance and generalization of the baselines.

**Training throughput.** Figure 5f shows the training throughput of the implemented baseline on meta-RL tasks. We used a 9x9 grid size and the `trivial-1m` benchmark with fixed model size and

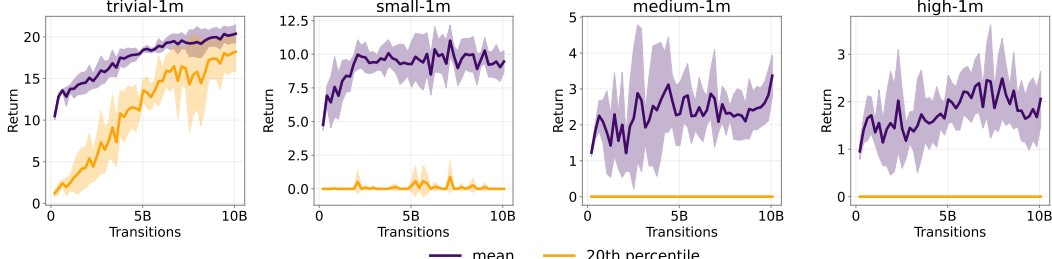

Figure 6: Learning curves from training an RL$^2$ (Duan et al., 2016; Wang et al., 2016) agent based on recurrent PPO (Schulman et al., 2017) on the XLand-MiniGrid meta-RL benchmarks introduced in Section 3. Grid size of 13x13 with four rooms was used (see Figure 14 in Appendix I for a visualization). We report the return for 25 trials on each evaluation task, 4096 tasks in total. During training, the agent is only limited by the fixed maximum step count and can get more trials if it manages to solve tasks faster. Results are averaged over five random seeds. One can see that the proposed benchmarks present a significant challenge, especially in terms of the 20th score percentile. Similar to Team et al. (2023), we evaluate algorithms mainly on the 20th percentile, as it better reflects the ability to adapt to new tasks. Also, see Appendix H for the learning curves on image-based observations.

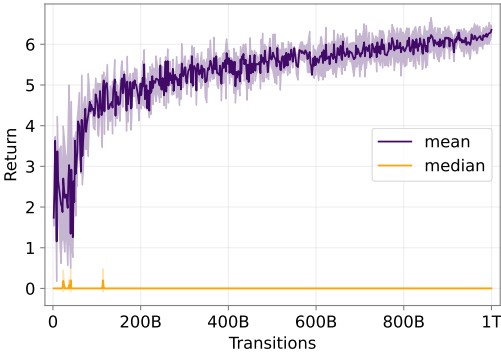

Figure 7: Learning curve from the large-scale baseline training on the `high-1m` benchmark for **one trillion** transitions. We use the same setup as in Figure 6 and report the return for 25 trials. Results are averaged over three random training seeds. While such an extreme amount of experience can be collected with our library, this alone is not enough to significantly improve the baseline performance on this benchmark.

Figure 8: Preliminary generalization evaluation of the implemented RL$^2$ Duan et al. (2016); Wang et al. (2016) baseline. For training, we used the same setup as in Figure 6, excluding a subset of goal types (but not rules) from the benchmarks, and then tested generalization on 4096 tasks sampled from them. We report the return for the 25 trials. Results are averaged over three random training seeds. A large gap in generalization remains, even for the benchmarks with minimal diversity.

RNN sequence length of 256. Although it is not always optimal to maximize the number of steps per second, as this may reduce the sample efficiency (Andrychowicz et al., 2020), for demonstration purposes we searched over other parameters to maximize per device utilization. As one can see, on a single device, the training throughput saturates near one million steps per second. Similar to Figure 5a, it can be greatly increased with multiple devices to almost reach six million. Further improvement may require changing the RNN architecture to a more hardware-friendly one, e.g., the Transformer (Vaswani et al., 2017), which we left for future work. We provide additional throughput benchmarks on consumer grade GPUs in the Appendix F.

**Training Performance.** We trained the implemented RL$^2$ on the proposed benchmarks to demonstrate that they provide an interesting challenge and to establish a starting point for future research (see Figure 6). A grid size of 13x13 with four rooms was used (see Figure 14). We report the return for 25 trials, averaged over all 4096 evaluation tasks and five random training seeds. During the training, the

agent was limited to a fixed number of steps in each task, so that it could get more trials if it was able to solve the tasks faster. Similar to (Team et al., 2023), we evaluate algorithms mainly on the 20th percentile, as it provides a lower bound that better reflects the ability to adapt to new tasks, while the average score is dominated by easy tasks for which it is easier to generalize. Therefore, we advise practitioners to compare their algorithms according to the 20th percentile. See appendix K for a more in-depth discussion.

As one can see, the performance is far from optimal, especially in terms of the 20th score percentile. Furthermore, the performance on `high-1m` remains sub-optimal even when the agent is trained for extreme **one trillion** transitions (see Figure 7). Note that training Ada (Team et al., 2023) for such a duration would not be feasible.[7]

**Generalization.** Since the main purpose of meta-learning is enabling rapid adaptation to novel tasks, we are mainly interested in generalization during training. To test the baseline along this dimension, we excluded a subset of goal types (but not rules) from the benchmarks and retrained the agent using the same hyperparameters and procedures as in Figure 6. During training, we tested generalization on the 4096 tasks sampled from the excluded (i.e., unseen) tasks and averaged over three random training seeds. As Figure 8 shows, there remains a large gap in generalization on all benchmarks, even on the one with minimal diversity.

## 5 Related Work

**Meta-learning environments.** Historically, meta-RL benchmarks have focused on tasks with simple distributions, such as bandits and 2D navigation (Wang et al., 2016; Duan et al., 2016; Finn et al., 2017), or few-shot control policy adaptation (e.g., MuJoCo (Zintgraf et al., 2019) or MetaWorld (Yu et al., 2020)), where the latent component is reduced to a few parameters that control goal location or changes in robot morphology. The recent wave of in-context reinforcement learning research (Laskin et al., 2022; Lee et al., 2023; Norman & Clune, 2023; Kirsch et al., 2023; Sinii et al., 2023; Zisman et al., 2023) also uses simple environments for evaluation, such as bandits, navigational DarkRoom & KeyToDoor, or MuJoCo with random projections of observations (Lu et al., 2023; Kirsch et al., 2023). Notable exceptions include XLand (Team et al., 2021, 2023) and Alchemy (Wang et al., 2021). However, XLand is not open source, while Alchemy is built on top of Unity (www.unity.com) and runs at 30 FPS, which is not enough for quick experimentation with limited resources.

We hypothesize that the popularity of such simple benchmarks can be attributed to their affordability, as meta-training requires significantly more environmental transitions than traditional single-task RL Team et al. (2023). However, being limited to simple benchmarks prevents researchers from uncovering the limits and scaling properties of the proposed methods. We believe that the solution to this is an environment that does not compromise interestingness and task complexity for the sake of affordability. Therefore, we designed XLand-MiniGrid to include the best from the XLand and Alchemy environments without sacrificing speed and scalability thanks to the JAX (Bradbury et al., 2018) ecosystem.

**Hardware-accelerated environments.** There are several approaches to increasing the throughput of environment experience. The most common approach would be to write the environment logic in low level languages (to bypass Python GIL) for asynchronous collection, as EnvPool Weng et al. (2022) does. However, this does not remove the bottleneck of data transfer between CPU and GPU on every iteration, and the difficulties of debugging asynchronous systems. Porting the entire environment to the GPU, as was done in Isaac Gym (Makoviychuk et al., 2021), Megaverse (Petrenko et al., 2021) or Madrona (Shacklett et al., 2023), can remove this bottleneck, but has the disadvantage of being GPU-only.

Recently, new environments written entirely in JAX Bradbury et al. (2018) have appeared, taking advantage of the GPU or TPU and the ability to compile the entire training loop just-in-time, further reducing the overall training time. However, most of them focus on single-task environments for robotics (Freeman et al., 2021), board games (Koyamada et al., 2023) or combinatorial optimization (Bonnet et al., 2023). The most similar to our work is Craftax (Matthews et al., 2024). It provides much more complex and varied world mechanics, but lacks the flexibility to customize the possible

---

[7]According to Appendix D.2 from Team et al. (2023), training Ada for 25B transitions took over 1 week on 64 Google TPUv3.

challenges or tasks which is not really suitable for meta-RL research. While in XLand-MiniGrid, thanks to a system of rules and goals, the user can generate many unique tasks of various difficulty.

**Grid World Environments.** Grid world environments have a long history in RL research Sutton & Barto (2018), as they offer a number of attractive properties. They are typically easy to implement, do not require large computational resources, and have simple observation spaces. However, they pose a significant challenge even to modern RL methods, making it possible to test exploration (Zhang et al., 2020), language understanding and generalization (Hanjie et al., 2021; Zholus et al., 2022; Lin et al., 2023; Chevalier-Boisvert et al., 2023), as well as memory (Paischer et al., 2022).

Despite the great variety of benefits of the existing grid world benchmarks, to the best of our knowledge, only the no longer maintained KrazyWorld (Stadie et al., 2018) focuses on meta-learning. Other libraries, such as the popular MiniGrid (Chevalier-Boisvert et al., 2023) and Griddly (Bamford et al., 2020), are not scalable and extensible enough to cover meta-learning needs. In this work, we have attempted to address these needs with new minimalistic MiniGrid-style grid world environments that can scale to millions of steps per second on a single GPU.

**Large-batch RL.** Large batches are known to be beneficial in deep learning (You et al., 2019) and deep reinforcement learning is no exception. It is known to be extremely sample-inefficient and large batches can increase the throughput, speeding up convergence and improving training stability. For example, many of the early breakthroughs on the Atari benchmark were driven by more efficient distributed experience collection (Horgan et al., 2018; Espeholt et al., 2018; Kapturowski et al., 2018), eventually reducing training time to just a few minutes (Stooke & Abbeel, 2018; Adamski et al., 2018) per game. Increasing the mini-batch size can also be beneficial in offline RL (Nikulin et al., 2022; Tarasov et al., 2023).

However, not all algorithms scale equally well, and off-policy methods have until recently lagged behind (Li et al., 2023), whereas on-policy methods, while generally less sample efficient, can scale to enormous batch sizes of millions (Berner et al., 2019; Petrenko et al., 2020) and complete training much faster (Stooke & Abbeel, 2018; Shacklett et al., 2021) in wall clock time. While we do not introduce any novel algorithmic improvements in our work, we hope that the proposed highly scalable XLand-MiniGrid environments will help practitioners perform meta-reinforcement learning experiments at scale faster and with fewer resources.

# 6 Conclusion

Unlike other RL subfields, meta-RL lacked environments that were both non-trivial and computationally efficient. To fill this gap, we developed XLand-MiniGrid, a JAX-accelerated library of grid-world environments and benchmarks for meta-RL research. Written in JAX, XLand-MiniGrid is designed to be highly scalable and capable of running on GPU or TPU accelerators, and can achieve millions of steps per second. In addition, we have implemented easy-to-use baselines and provided preliminary analysis of their performance and generalization, showing that the proposed benchmarks are challenging. We hope that XLand-MiniGrid will help democratize large-scale experimentation with limited resources, thereby accelerating meta-RL research.

## Acknowledgments and Disclosure of Funding

At the time of writing, all authors were employed by T-Bank or/and AIRI. All computational resources were provided by T-Bank and AIRI.

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

## A   General Ethic Conduct and Potential Negative Societal Impact

To the best of our knowledge, our work does not have a direct potential negative impact on society. On the contrary, by democratizing experimentation, fewer resources can be used to conduct large-scale studies reducing the carbon emissions produced by the GPU accelerators.

## B   License

XLand-MiniGrid is open-source and available at `https://github.com/corl-team/xland-minigrid` under Apache 2.0 license.

## C   Limitations and Future Work

There are several notable limitations to our work, some of which we expect to address in future releases of our library.

Compared to the full-scale XLand (Team et al., 2021, 2023), we do not currently support multi-agent simulations, procedural generation of complex worlds, rules with multiple output entities, or goal composition. Our initial environments use simple multi-room grid worlds (see Section 2.3), although we plan to add procedural generation of different maze layouts. Full-scale XLand also provides 30 single-agent probe tasks that were designed by hand to be interpretable and intuitive to the humans. In contrast, we evaluated our baselines on tasks sampled from the provided benchmarks.

Compared to the MiniGrid (Chevalier-Boisvert et al., 2023), we do not yet support all tiles such as lava or moving obstacles. Also, as our focus is on meta-RL, we do not provide explicit natural language encoding of rules and goals, although this can easily be done. For a language focused learning environment, we refer the reader to the recent HomeGrid (Lin et al., 2023) environment. Although we currently do not support all existing MiniGrid environments, we provide enough tools to make it easy to implement them if needed (and have demonstrated this in the Section 2.3).

In general, while JAX is much more high-level than CUDA or Triton (Tillet et al., 2019), it is still much more restrictive than PyTorch (Paszke et al., 2019), can be difficult to debug, and is poorly suited to the heterogeneous computations or conditional branching that are common when implementing environments. However, as we show in our work, when used correctly it can provide excellent scalability opportunities.

## D  Additional Code Examples

We provide additional code examples for extended environment usage with benchmarks and jit-compilation of the entire environment rollout.

```python
import jax.random
import xminigrid
from xminigrid.benchmarks import Benchmark

# to list available benchmarks: xminigrid.registered_benchmarks()
# downloading to path specified by XLAND_MINIGRID_DATA,
# ~/.xland_minigrid by default
benchmark: Benchmark = xminigrid.load_benchmark(name="trivial-1m")
# reusing cached on the second use
benchmark: Benchmark = xminigrid.load_benchmark(name="trivial-1m")

# users can sample or get specific rulesets
benchmark.sample_ruleset(jax.random.PRNGKey(0))
benchmark.get_ruleset(ruleset_id=benchmark.num_rulesets() - 1)

# or split them for train & test
train, test = benchmark.shuffle(key=jax.random.PRNGKey(0)).split(prop=0.8)

# usage with the environment
key = jax.random.PRNGKey(0)
reset_key, ruleset_key = jax.random.split(key)

benchmark = xminigrid.load_benchmark(name="trivial-1m")
# choosing a ruleset, see section on rules and goals
ruleset = benchmark.sample_ruleset(ruleset_key)

# to list available environments: xminigrid.registered_environments()
env, env_params = xminigrid.make("XLand-MiniGrid-R9-25x25")
env_params = env_params.replace(ruleset=ruleset)

# fully jit-compatible step and reset methods
timestep = jax.jit(env.reset)(env_params, reset_key)
timestep = jax.jit(env.step)(env_params, timestep, action=0)

# optionally render the state
env.render(env_params, timestep)
```

Listing 2: Extended usage example (see Listing 1) showcasing how to load benchmarks, sample rulesets and combine them with environments.

```python
import jax
import jax.tree_utils as jtu
import xminigrid
from xminigrid.wrappers import GymAutoResetWrapper

# alternatively, users can provide step_fn and reset_fn instead
# of closure, but this way is simpler to use after creation
def build_rollout(env, env_params, num_steps):
    def rollout(rng):
        def _step_fn(carry, _):
            rng, timestep = carry
            rng, _rng = jax.random.split(rng)
            action = jax.random.randint(
                _rng,
                shape=(),
                minval=0,
                maxval=env.num_actions(env_params)
            )
            timestep = env.step(env_params, timestep, action)
            return (rng, timestep), timestep

        rng, _rng = jax.random.split(rng)

        timestep = env.reset(env_params, _rng)
        rng, transitions = jax.lax.scan(
            _step_fn, (rng, timestep), None, length=num_steps
        )
        return transitions

    return rollout

env, env_params = xminigrid.make("MiniGrid-EmptyRandom-8x8")
# do not forget to use auto reset wrapper!
env = GymAutoResetWrapper(env)

# jiting the entire rollout
rollout_fn = jax.jit(build_rollout(env, env_params, num_steps=1000))

# first execution will compile
transitions = rollout_fn(jax.random.PRNGKey(0))

print("Transitions shapes: \n", jtu.tree_map(jnp.shape, transitions))
```

Listing 3: Example code on how to jit-compile the entire rollout through the episodes. This can be further easily parallelized with jax.vmap over a batch of environments or random seeds.

# E  Additional Benchmark Figures

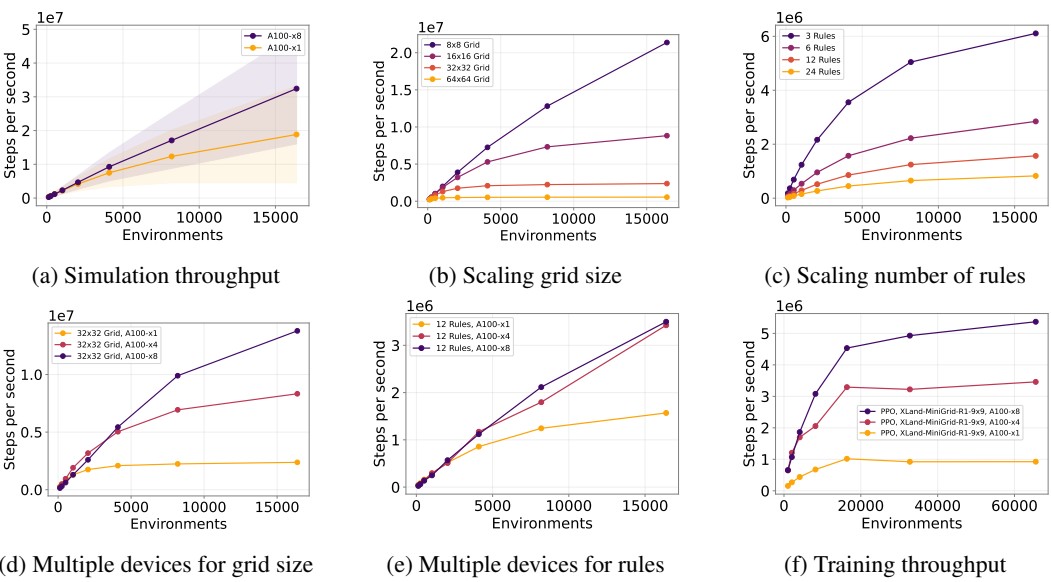

Figure 9: Figures analogous to Figure 5 from Section 4.1, but without the log axes, to provide an alternative visual representation. We hope they will help practitioners to better understand the actual scaling behavior. All hyperparameters and settings other than axis scaling are identical to Figure 5.

# F  Additional Benchmarks on Consumer Grade GPUs

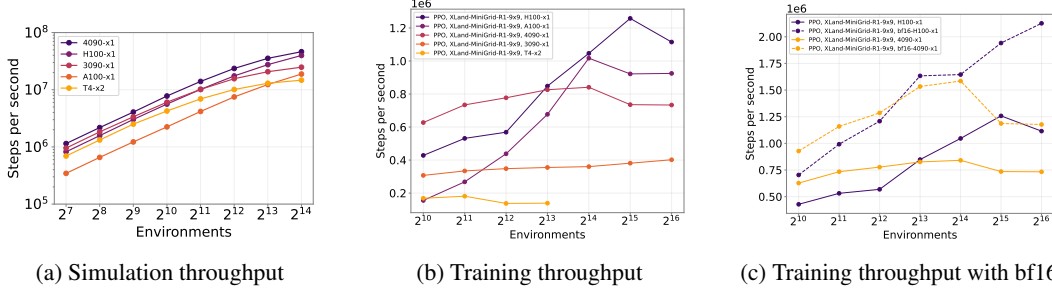

Figure 10: Additional simulation and training throughput benchmarks on consumer grade GPUs.

While our main experiments and benchmarks were done on A100 and H100 GPUs, we further tried to complement them with results on more affordable consumer GPUs like 4090, 3090 or freely available T4s from Google Colab or Kaggle Notebooks (see Figure 10). As can be seen on Figure 10a, pure throughput with random policy is decent on all GPUs, even the oldest T4 is capable of achieving millions of steps per second (SPS) and scales with more environments.

However, during PPO training the difference becomes more apparent (see Figure 10b). The maximum for a T4x2 is around 160k and for 3090 is around 400k and for 4090 is around 800k SPS. To compare, A100 is able to achieve 1.0M and H100 even higher 1.2M steps per second. The most important factor during training is how fast we can get through a single epoch on the batch collected from all environments, which in turn comes down to the maximum mini-batch size. Ideally, for every increase in the number of parallel environments, we should also increase the mini-batch size. If this does not happen, saturation occurs. Due to the smaller memory capacity of consumer GPUs the saturation during training comes a lot earlier than for A100/H100. However, 800k on 4090 is still a good result, more than any other benchmark (like MetaWorld, Alchemy, etc) can achieve with such resources.

To make training more accessible on the next generation of GPUs (including consumer ones), we added support for bf16 precision during training. On our hardware and with same hyperparamers it increased throughput dramatically on H100 and 4090 GPUs, reaching 1.5M and 2.1M respectively (see Figure 10c).

## G   Experiments With Goal Conditioning

In contrast to the (Team et al., 2023), in our main experiments we kept the rules and goals hidden from the agent to test baselines in the pure meta-learning setting, where all necessary information about the task should be uncovered through intelligent exploration. However, this is our conscious choice, not a limitation of the proposed environment, and we can easily provide this information to the agent if needed. To illustrate this, we conducted a preliminary experiment in the multitask goal-conditioned setting (see Figure 11). To encode rules and a goal, we pre-embed them and concatenate all resulting embeddings to obtain the final representation. We then condition the agent on this representation, concatenating it with an image representation after the CNN and before the RNN. We train the agent simultaneously on the 65536 tasks from the `medium-1m` benchmark and evaluate it on the 1024 new tasks. While the agent can generalize on simple rulesets and solve them, albeit not optimally, in the end, in general there is still a lot of room for improvement, especially in terms of generalization to harder tasks (near zero 20th percentile).

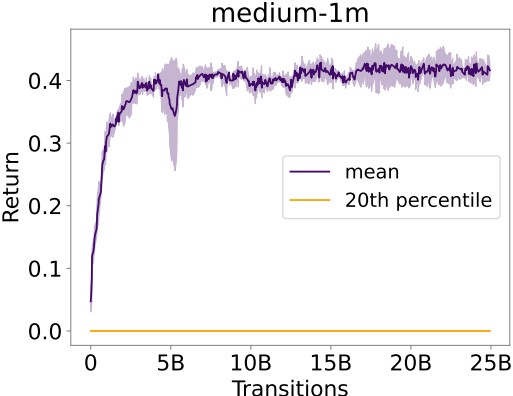

Figure 11: Learning curve of a multitask goal-conditioned agent based on the recurrent PPO. We simultaneously trained the agent on the 65536 tasks from the `medium-1m` benchmark and evaluated it on the 1024 new tasks. While the agent can generalize to simple rulesets and solve them, albeit not optimally, in the end there is generally still a lot of room for improvement, especially in terms of generalization to harder tasks (near zero 20th percentile). All other hyperparameters were taken from the single-task PPO experiments.

## H   Experiments With Image Observations

Although symbolic observations are efficient and fast, they can be limiting, as they greatly simplify perception. They also make generalization much more difficult, as there are often no trained embeddings for new objects. Furthermore, symbolic observations may be too small to be used with some large pre-trained visual models. In general, we don't feel that these limitations are severe enough to sacrifice efficiency, but we recognise that practitioners should be able to work around them if they need to. Therefore, in addition to the efficient encoding of symbolic observations using tile and color IDs as in the MiniGrid (described in Section 2.2), we provide an alternative way of encoding symbolic observations by rendering them as $224 \times 224$ RGB images using a wrapper (see Listing 4 for a usage example).

We have replicated some of the experiments from Section 4 with the image observations, confirming that they do indeed make benchmarks harder (see Figure 12), but can introduce large overheads that significantly reduce simulation (Figure 13) and training throughput. The pure environment throughput

can still be on the order of millions of steps per second. Unfortunately, during training we can fit far fewer parallel environments (only 1024) on a single GPU due to the increased memory consumption of the larger model and images. As a result, we got about 20k steps per second during training, which makes it too long ($\sim 8$ hours) to train models with more than 250M transitions on a single GPU.

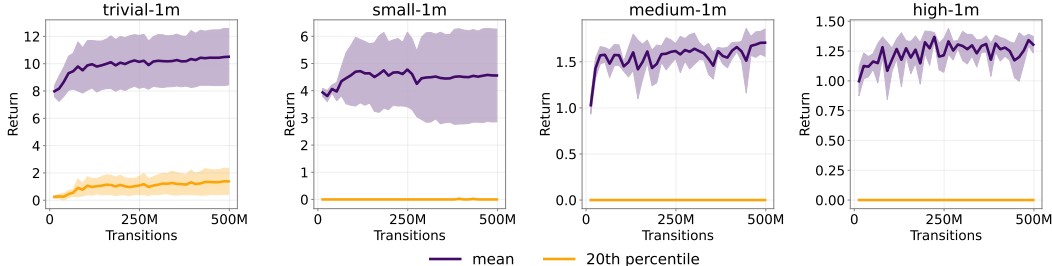

Figure 12: Learning curves from training an RL$^2$ (Duan et al., 2016; Wang et al., 2016) agent based on recurrent PPO (Schulman et al., 2017) on the XLand-MiniGrid meta-RL benchmarks introduced in Section 3. Compared to the Figure 6 we use image observations instead of the symbolic ones. We also a slightly larger CNN to handle larger images. Grid size of 13x13 with four rooms was used (see Figure 14 in Appendix I for a visualization). We report the return for 25 trials on each evaluation task, 4096 tasks in total. During training, the agent is only limited by the fixed maximum step count and can get more trials if it manages to solve tasks faster. Results are averaged over five random seeds. One can see that the proposed benchmarks present a significant challenge, especially in terms of the generalization, as 20th percentile is near zero.

```python
import jax
import xminigrid
from xminigrid.experimental.img_obs import RGBImgObservationWrapper

key = jax.random.key(0)

env, env_params = xminigrid.make("XLand-MiniGrid-R9-25x25")

assert (
env.observation_shape(env_params) ==
 (env_params.view_size, env_params.view_size, 2)
)

# for faster rendering, pre-rendered tiles will
# be saved at XLAND_MINIGRID_CACHE path
# use XLAND_MINIGRID_RELOAD_CACHE=True to force cache reload
env = RGBImgObservationWrapper(env)

# observation is RGB image now!
assert env.observation_shape(env_params) == (224, 224, 3)

# jitting works as usuall
timestep = jax.jit(env.reset)(env_params, reset_key)
timestep = jax.jit(env.step)(env_params, timestep, action=0)
```

Listing 4: Example code on how to render symbolic observations as RGB images.

## I  Library Details

After the release, the library will be made available as a package on PyPI. For efficiency, we separate rules and goals with the same meaning that apply only to objects or to the agent, since the agent is not considered a valid entity internally. For example, the production rules AgentNear and TileNear are separated while having very similar effects. Similarly, due to the requirement to be

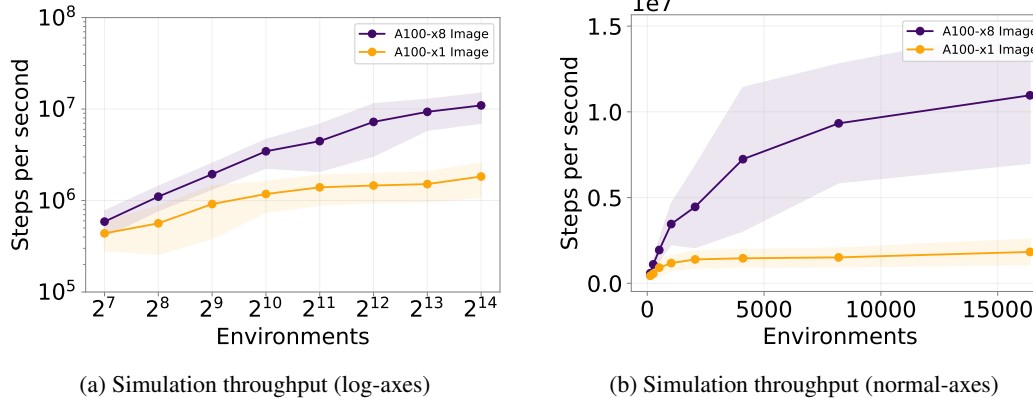

| (a) Simulation throughput (log-axes) | (b) Simulation throughput (normal-axes) |

Figure 13: Simulation throughput averaged over all registered environment variations (38 in total, see Section 2.3) using image observations instead of symbolic ones. Although the throughput is noticeably lower compared to Figure 5a, we can still achieve millions of steps per second on a single GPU and even more on multiple GPUs.

compatible with jit-compilation, new rules and goals can not be easily added by users, without cloning the repository and modifying the source code, as they are hard-coded in `jax.lax.switch` inside the `xminigrid.core.rules.check_rule` and `xminigrid.core.goals.check_goal` functions. We hope that the diversity provided by already available rules and goals will be enough for most common use cases. We list all supported objects in Table 1, and all rules and goals with their descriptions in Tables 2 and 3.

To further increase diversity, we implemented several common grid layouts with multiple rooms (see Figure 14), inspired by MiniGrid. We provide layouts with 1, 2, 4, 6 and 9 rooms. Positions of the doors, as well as their colors, and objects are randomized between resets, but the grid itself does not change. Currently, they should be chosen in advance and can not be changed under jit-compilation. This restriction can be relaxed, but will have high overhead during reset, as due to the branching and under `jax.vmap` all conditional branches will be evaluated. We left efficient procedural grid generation for future work.

Table 1: Supported Objects Types

(a) Tiles

| Tile | ID |
|---|---|
| END_OF_MAP | 0 |
| UNSEEN | 1 |
| EMPTY | 2 |
| FLOOR | 3 |
| WALL | 4 |
| BALL | 5 |
| SQUARE | 6 |
| PYRAMID | 7 |
| GOAL | 8 |
| KEY | 9 |
| DOOR_LOCKED | 10 |
| DOOR_CLOSED | 11 |
| DOOR_OPEN | 12 |
| HEX | 13 |
| STAR | 14 |

(b) Colors

| Color | ID |
|---|---|
| END_OF_MAP | 0 |
| UNSEEN | 1 |
| EMPTY | 2 |
| RED | 3 |
| GREEN | 4 |
| BLUE | 5 |
| PURPLE | 6 |
| YELLOW | 7 |
| GREY | 8 |
| BLACK | 9 |
| ORANGE | 10 |
| WHITE | 11 |
| BROWN | 12 |
| PINK | 13 |

Table 2: Supported Goals.

| Goal | Meaning | ID |
|------|---------|-----|
| `EmptyGoal` | Placeholder goal, always returns False | 0 |
| `AgentHoldGoal(a)` | Whether agent holds `a` | 1 |
| `AgentOnTileGoal(a)` | Whether agent is on tile `a` | 2 |
| `AgentNearGoal(a)` | Whether agent and `a` are on neighboring tiles | 3 |
| `TileNearGoal(a, b)` | Whether `a` and `b` are on neighboring tiles | 4 |
| `AgentOnPositionGoal(x, y)` | Whether agent is on `(x, y)` position | 5 |
| `TileOnPositionGoal(a, x, y)` | Whether `a` is on `(x, y)` position | 6 |
| `TileNearUpGoal(a, b)` | Whether `b` is one tile above `a` | 7 |
| `TileNearRightGoal(a, b)` | Whether `b` is one tile to the right of `a` | 8 |
| `TileNearDownGoal(a, b)` | Whether `b` is one tile below `a` | 9 |
| `TileNearLeftGoal(a, b)` | Whether `b` is one tile to the left of `a` | 10 |
| `AgentNearUpGoal(a)` | Whether `a` is one tile above agent | 11 |
| `AgentNearRightGoal(a)` | Whether `a` is one tile to the right of agent | 12 |
| `AgentNearDownGoal(a)` | Whether `a` is one tile below agent | 13 |
| `AgentNearLeftGoal(a)` | Whether `a` is one tile to the left of agent | 14 |

Table 3: Supported Rules.

| Rule | Meaning | ID |
|------|---------|-----|
| `EmptyRule` | Placeholder rule, does not change anything | 0 |
| `AgentHoldRule(a) → c` | If agent holds `a` replaces it with `c` | 1 |
| `AgentNearRule(a) → c` | If agent is on neighboring tile with `a` replaces it with `c` | 2 |
| `TileNearRule(a, b) → c` | If `a` and `b` are on neighboring tiles, replaces one with `c` and removes the other | 3 |
| `TileNearUpRule(a, b) → c` | If `b` is one tile above `a`, replaces one with `c` and removes the other | 4 |
| `TileNearRightRule(a, b) → c` | If `b` is one tile to the right of `a`, replaces one with `c` and removes the other | 5 |
| `TileNearDownRule(a, b) → c` | If `b` is one tile below `a`, replaces one with `c` and removes the other | 6 |
| `TileNearLeftRule(a, b) → c` | If `b` is one tile to the left of `a`, replaces one with `c` and removes the other | 7 |
| `AgentNearUpRule(a) → c` | If `a` is one tile above agent, replaces it with `c` | 8 |
| `AgentNearRightRule(a) → c` | If `a` is one tile to the right of agent, replaces it with `c` | 9 |
| `AgentNearDownRule(a) → c` | If `a` is one tile below agent, replaces it with `c` | 10 |
| `AgentNearLeftRule(a) → c` | If `a` is one tile to the left of agent, replaces it with `c` | 11 |

# J    Benchmarks Details

We provide scripts used to generate benchmarks described in Section 3 along with the main library, at `scripts/ruleset_generator.py` and `scripts/generate_benchmarks.sh`. We did not aim to make the generator fast, as benchmarks are rarely updated, so generating large numbers of tasks can take a long time (about 5+ hours, with a lot of time spent filtering out repeated tasks). However, we tried to make it reproducible with a fixed random seed. For exact parameters used, see Table 4. We document the meaning of each parameter in the corresponding script in the code.

During generation, 10 colors were used (such as red, green, blue, purple, yellow, gray, white, brown, pink, orange), and 7 objects at most (such as ball, square, pyramid, key, star, hex, goal). This gives at most 70 unique objects to choose from during generation, and currently limits the maximum depth of the main task tree to 5, as in the worst-case scenario, it will be a full binary tree, where each node adds two new objects. Complexity can be further increased by more aggressive branch pruning to increase overall depth, or by including distractor rules. The disappearance production rule was emulated by setting the production tile to the black floor.

We also report benchmark sizes in Table 5. As stated in Section 3, they are hosted in the cloud in a compressed format and are automatically downloaded and cached on first use (by default to the path specified in `$XLAND_MINIGRID_DATA` variable). Additionally, we provide all the tools needed for users to load new benchmarks generated from the script with custom parameters (with the `xminigrid.load_benchmark_from_path`). In the compressed format, all benchmarks take up less than 100MB, which will be uncompressed and loaded into the memory during loading. By default, JAX will load them into the accelerator memory if available, which can be problematic, as 600MB of GPU memory can be a lot. For memory-constrained devices (such as the RTX 3060 or similar), we advise explicitly putting benchmarks on CPU memory, using the standard tools in JAX

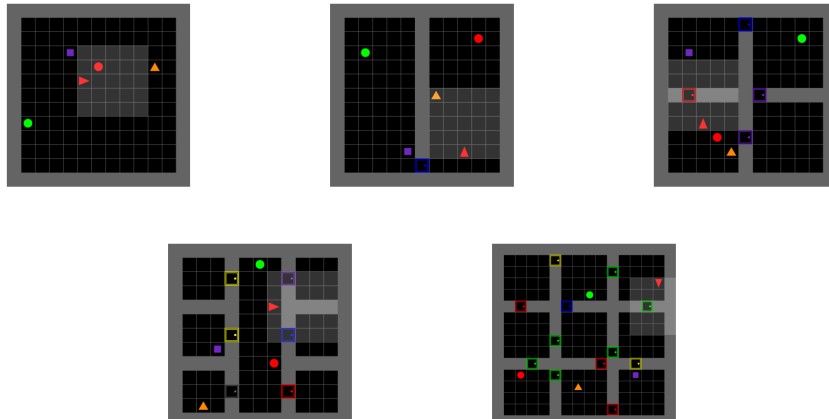

Figure 14: Visualization of the available layouts in XLand-MiniGrid (see Section 2.3). We provide layouts with 1, 2, 4, 6 and 9 rooms. The overall grid size can be changed. Positions of the doors (as well as their colors) and objects are randomized between resets, but the grid itself does not change (on the grid with 6 rooms, the doors are also fixed). Currently, they should be chosen in advance and can not be modified under jit-compilation. This restriction can be changed, but will have high overhead during reset, as due to the branching and under `jax.vmap` all conditional branches will be evaluated (i.e., all grids will be generated and then only one selected). We left efficient procedural grid generation for future work.

(e.g., `jax.device_put`). However, in such cases, there may be overhead due to transfer from CPU memory to GPU memory during training.

Table 4: Parameters used for the generation of benchmarks proposed in this work (see Section 3). The argument names are exactly the same as the arguments of the script used to generate them. The script is provided along with the library in `scripts/ruleset_generator.py` and `scripts/generate_benchmarks.sh`. We tried to ensure that the generation will be reproducible with the same random seed.

| Parameter | trivial | small | medium | high |
|---|---|---|---|---|
| chain_depth | 0 | 1 | 2 | 3 |
| sample_depth | false | false | false | false |
| prune_chain | false | true | true | true |
| prune_prob | 0.0 | 0.3 | 0.1 | 0.1 |
| num_distractor_rules | 0 | 2 | 3 | 4 |
| sample_distractor_rules | false | true | true | true |
| num_distractor_objects | 3 | 2 | 2 | 1 |
| random_seed | 42 | 42 | 42 | 42 |

## K   Experiment Details

Below, we provide additional details about each experiment from Section 4.2.

**Training throughput.** As noted in Section 4.2, we used the `XLand-MiniGrid-R1-9x9` environment and the `trivial-1m` benchmark. Overall, we used the default hyperparameters that are common in other PPO implementations, such as CleanRL (Huang et al., 2022) and PureJaxRL (Lu et al., 2022). After a small sweep on model size and RNN sequence length, we took the best performing ones and fixed them (see Table 6). Next, when measuring the training throughput, we tuned `num_envs` and `num_minibatches` to maximize the per-device utilization for each setup, which we define as a saturation point, after which further increasing the minibatch size or the number of parallel environments does not increase the overall throughput. Note that the values obtained are close to

Table 5: Sizes of the benchmarks provided by XLand-MiniGrid. Benchmarks are stored in a compressed state and will be downloaded from the cloud and cached locally on first use. During loading, they will be uncompressed and loaded into memory. By default, JAX will load them into the accelerator memory if available. For devices limited in memory, we advise explicitly putting benchmarks on CPU memory by using the standard tools in JAX (e.g. `jax.device_put`). Note that this can introduce some overhead during training.

| Benchmark | Size (MB) | Comp. Size (MB) |
|-----------|-----------|-----------------|
| trivial-1m | 38.0 | 5.7 |
| small-1m | 69.0 | 13.7 |
| medium-1m | 112.0 | 17.7 |
| medium-3m | 336.0 | 53.1 |
| high-1m | 193.0 | 31.6 |
| high-3m | 579.0 | 94.8 |

Table 6: RL$^2$ hyperparameters used in experiments from Section 4.2. Most of them were ported from CleanRL (Huang et al., 2022) or PureJaxRL (Lu et al., 2022).

| Parameter | Value |
|-----------|-------|
| action_emb_dim | 16 |
| rnn_hidden_dim | 1024 |
| rnn_num_layers | 1 |
| head_hidden_dim | 256 |
| num_envs | 16384 |
| num_steps_per_env | 12800 |
| num_steps_per_update | 256 |
| update_epochs | 1 |
| num_minibatches | 32 |
| total_timesteps | 1e10 |
| optimizer | Adam |
| lr | 0.001 |
| clip_eps | 0.2 |
| gamma | 0.99 |
| gae_lambda | 0.95 |
| ent_coef | 0.01 |
| vf_coef | 0.5 |
| max_grad_norm | 0.5 |
| eval_num_envs | 4096 |
| eval_num_episodes | 25 |
| eval_seed | 42 |
| train_seed | 42 |

the maximum possible on our hardware (A100 with full precision). Throughput will decrease as the complexity of the benchmark or the size of the agent network increases.

**Performance.** All the main performance experiments (see Figure 6) were performed with `XLand-MiniGrid-R4-13x13` and a single A100 GPU at full precision. We used the same hyperparameters from Table 6 for all benchmarks. Depending on the benchmark, training to 10B transitions took between 3 and 5 hours. For the large-scale experiment (see Figure 7), we used the `high-1m` benchmark, increasing the number of devices from one to eight A100s and the number of environments from 16384 to 131072 (simply multiplying by eight, as it was close to optimal for a single device). Training to the 1T transitions took 72 hours or 3 days, with a throughput of $\sim 3.85$M steps per second.

In our main experiments, we used the 20th percentile to evaluate our baselines, following the approach of Team et al. (2023). Unlike the average score, which can be dominated by outlier scores from the easiest tasks, the 20th percentile gives us a lower bound guarantee of the agent's ability to adapt to a majority of tasks from the broad evaluation distribution. For example, if an agent's 20th percentile score is 0.9, then the agent will score at least 0.9 on 80% of the evaluation tasks. Since our main goal

with meta-RL is to learn an agent that can adapt to new, unseen tasks, we believe that maximizing the lower bound on performance is the most appropriate approach.

**Generalization.** To assess generalization in Figure 8, we excluded a whole subset of goal types from the benchmarks instead of just splitting the tasks to increase the novelty of the test tasks. During training, only goals with IDs 1, 3, 4 were retained, while all others were separated into the test from which we sampled 4096 tasks. After the split, there were $\sim 300$k training tasks left in each benchmark. The exclusion of some rule types will likely further increase the generalization gap. However, more aggressive filtering will probably require the use of benchmarks with a larger total number of tasks, e.g., variants with 3M of tasks.

# L Environments

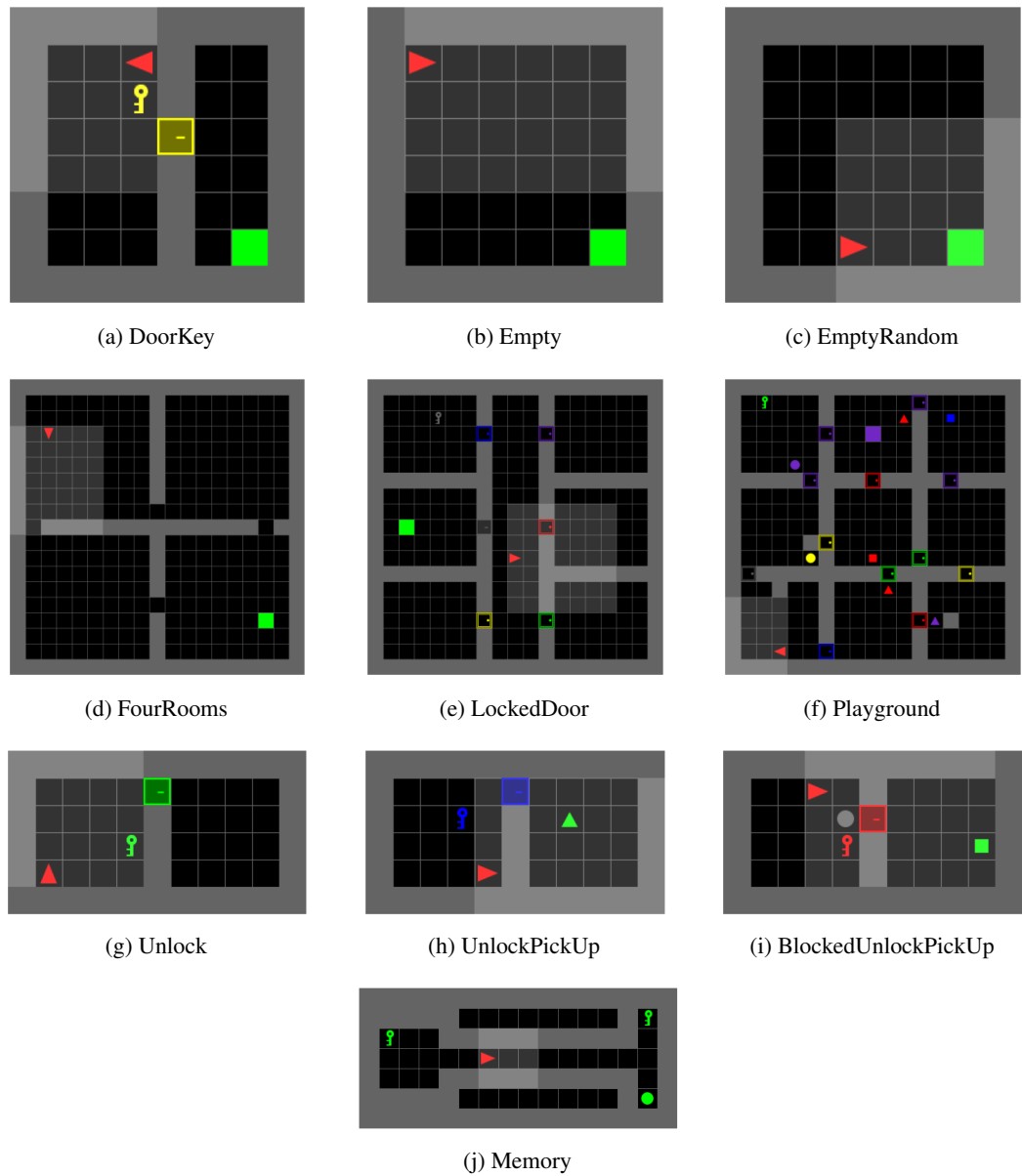

(a) DoorKey  (b) Empty  (c) EmptyRandom

(d) FourRooms  (e) LockedDoor  (f) Playground

(g) Unlock  (h) UnlockPickUp  (i) BlockedUnlockPickUp

(j) Memory

Figure 15: Visualization of all environments ported from the original MiniGrid. Each can have different registered configurations (see Table 7). For a full description of each environment, we refer the reader to the MiniGrid documentation (`https://minigrid.farama.org`).

Table 7: Full list of the registered XLand-MiniGrid environments.

| Environment Name |
| --- |
| XLand-MiniGrid-R1-9x9 |
| XLand-MiniGrid-R1-13x13 |
| XLand-MiniGrid-R1-17x17 |
| XLand-MiniGrid-R2-9x9 |
| XLand-MiniGrid-R2-13x13 |
| XLand-MiniGrid-R2-17x17 |
| XLand-MiniGrid-R4-9x9 |
| XLand-MiniGrid-R4-13x13 |
| XLand-MiniGrid-R4-17x17 |
| XLand-MiniGrid-R6-13x13 |
| XLand-MiniGrid-R6-17x17 |
| XLand-MiniGrid-R6-19x19 |
| XLand-MiniGrid-R9-16x16 |
| XLand-MiniGrid-R9-19x19 |
| XLand-MiniGrid-R9-25x25 |
| MiniGrid-BlockedUnlockPickUp |
| MiniGrid-DoorKey-5x5 |
| MiniGrid-DoorKey-6x6 |
| MiniGrid-DoorKey-8x8 |
| MiniGrid-DoorKey-16x16 |
| MiniGrid-Empty-5x5 |
| MiniGrid-Empty-6x6 |
| MiniGrid-Empty-8x8 |
| MiniGrid-Empty-16x16 |
| MiniGrid-EmptyRandom-5x5 |
| MiniGrid-EmptyRandom-6x6 |
| MiniGrid-EmptyRandom-8x8 |
| MiniGrid-EmptyRandom-16x16 |
| MiniGrid-FourRooms |
| MiniGrid-LockedRoom |
| MiniGrid-MemoryS8 |
| MiniGrid-MemoryS16 |
| MiniGrid-MemoryS32 |
| MiniGrid-MemoryS64 |
| MiniGrid-MemoryS128 |
| MiniGrid-Playground |
| MiniGrid-Unlock |
| MiniGrid-UnlockPickUp |

