# OpenReview forum: "XLand-MiniGrid: Scalable Meta-Reinforcement Learning Environments in JAX"
_NeurIPS.cc/2024/Datasets_and_Benchmarks_Track — NeurIPS 2024 Track Datasets and Benchmarks Poster_

### Official Review · Reviewer_vqDw · 2024-06-13

**Rating:** 5
**Confidence:** 5
**Correctness:** See above.
**Clarity:** See above.

**Review:**

## Originality
This paper addresses the gap in the meta-RL field by providing an open-source, high-quality, high-throughput benchmark suite. While there are influences from existing works, the combination and execution are original enough.

## Quality
The engineering effort behind this benchmark is commendable, achieving impressive throughput and scalability. However, the focus on throughput alone is insufficient. The paper lacks in-depth experiments to validate the benchmark's suitability for meta-RL, and it does not test additional baselines beyond RL^2, which limits the comprehensiveness of the benchmarking results. Some design choices, such as distractor objects and hidden goals, are not thoroughly investigated.

## Clarity
The writing is generally clear with good visual examples.

## Significance
The benchmark has the potential to benefit meta-RL research by providing a scalable and efficient environment. However, the paper needs a more detailed exploration of its effectiveness for meta-RL and a comparative analysis with existing benchmarks to fully establish its significance. The limited scope of experiments using only RL^2 and the lack of discussion on societal impacts and benchmark limitations are notable drawbacks.

## Pros
- Addresses a gap in meta-RL with an original benchmark suite.
- Achieves high throughput and scalability, making large-scale training feasible.
- Well-written with good visual examples and scholarly composition.

## Cons
- Lacks in-depth experiments to validate the benchmark's suitability for meta-RL.
- Limited experimental scope, with only RL^2 tested.
- Some statements and figures are confusing or misleading.
- Missing comparative analysis with existing meta-RL benchmarks.
- Insufficient discussion on societal impacts and benchmark limitations.

------
I acknowledge I've read authors' response. I'd like to maintain my original evaluation.

**Strengths:**

See above.

**Additional Feedback:**

See above.

**Documentation:**

Well-documented.

**Ethics:**

No ethics concern.

**Limitations:**

See above.

**Opportunities For Improvement:**

See above.

**Relation To Prior Work:**

See above.

**Summary And Contributions:**

The paper introduces XLand-MiniGrid, a suite of grid-world environments and tools designed for meta-reinforcement learning (meta-RL) research. It merges the simplicity of MiniGrid with the diversity of tasks found in XLand, drawing motivation from the latter. The environment generator and simulation performance are highlighted as being scalable, supporting large-scale experimentation while maintaining efficiency.

---

> ### Author Rebuttal · Authors · 2024-08-11
>
> We thank the reviewer for their valuable comments and concerns. As we prepare a more comprehensive response, we would be very grateful for some clarification.
>
> > Some statements and figures are confusing or misleading.
>
> Could you please point out specific statements and figures that you feel are misleading?
>
> > Insufficient discussion on societal impacts and benchmark limitations.
>
> We discuss the possible limitations of our benchmark (specifically compared to the original XLand and MiniGrid, and limitations in general) in Appendix B & C of the current version of the text. Do you find the current description inadequate? What would you like to see in addition to what there is now? Should we consider moving it to the main body of the paper?
>
> > Missing comparative analysis with existing meta-RL benchmarks.
>
> We describe and compare XLand-MiniGrid with the existing meta-RL benchmarks (such as MetaWorld, KeyToDoor & DarkRoom, Alchemy, XLand and others) in Appendix D of the current version of the text. Do you feel that these comparisons are incomplete? We would be happy to improve the current description, but we need a more specific specification of the currently existing gaps.

---

> > ### Author Rebuttal · Authors · 2024-08-16
> >
> > > Limited experimental scope, with only RL^2 tested.
> > >
> >
> > As our main contribution and focus is on fast and efficient environments, we do not aim to provide new insights into the variety of meta-learning methods, but rather to provide an accessible platform for research and a solid foundation for practitioners to start building on.
> >
> > This is a common pattern, and in fact many papers similar to ours [1, 2, 3, 4] do not provide exhaustive benchmarking beyond the most popular and simplest algorithms in the field of choice, which despite its age is usually still a variant of PPO. Even the currently unsolved Nethack [4] environment only provided PPO or IMPALA baselines, as PPO is considered by many in the community to be the most stable, performant and highly scalable algorithm, while remaining relatively simple. RL^2 is also, at its core, just PPO + memory, and thus inherits all the benefits of scale and stability over gradient-based counterparts [5, 6]. Moreover, some of the more recent meta-learning algorithms such as [7, 8, 9], despite having different names, essentially simply replace the memory backbone from the old LSTM with something new, without changing the learning algorithm itself, which is PPO and thus RL^2.
> >
> > Given that there are no libraries for meta-RL with tested algorithms like CleanRL, stable-baslines3, torchrl and others, and implementation of algorithms in meta-RL is much more complicated (especially so that it is compatible with jit in JAX), we decided that it is much more important to provide a single but high-quality and fast implementation of an algorithm for which there are evidences that it can solve very complex problems if sufficiently scaled up.
> >
> > Therefore, we believe that PPO based RL^2 is the most appropriate baseline choice for our environments, as it encompasses many of the newer algorithms and allows practitioners to start simple and iterate fast, without taking away their ability to quickly incorporate new advances when the time comes.
> >
> > > Lacks in-depth experiments to validate the benchmark's suitability for meta-RL.
> > >
> >
> > > Missing comparative analysis with existing meta-RL benchmarks
> > >
> >
> > > Insufficient discussion on societal impacts and benchmark limitations.
> > >
> >
> > > Some statements and figures are confusing or misleading.
> > >
> >
> > As for the other concerns, we feel that they are not formulated precisely enough for us to take any active steps to address them on top of what we have already tried to do in our responses to other reviewers. We look forward to receiving clarifications and are happy to do our best to improve our paper based on the new feedback.
> >
> >
> > References:
> >
> > 1. Bonnet, C., Luo, D., Byrne, D., Surana, S., Coyette, V., Duckworth, P., ... & Laterre, A. (2023). Jumanji: a diverse suite of scalable reinforcement learning environments in jax. *arXiv preprint arXiv:2306.09884*.
> > 2. Koyamada, S., Okano, S., Nishimori, S., Murata, Y., Habara, K., Kita, H., & Ishii, S. (2023). Pgx: Hardware-accelerated parallel game simulation for reinforcement learning. *arXiv preprint arXiv:2303.17503*.
> > 3. Balloch, J., Lin, Z., Hussain, M., Srinivas, A., Wright, R., Peng, X., ... & Riedl, M. (2022). Novgrid: A flexible grid world for evaluating agent response to novelty. *arXiv preprint arXiv:2203.12117*.
> > 4. Küttler, H., Nardelli, N., Miller, A., Raileanu, R., Selvatici, M., Grefenstette, E., & Rocktäschel, T. (2020). The nethack learning environment. *Advances in Neural Information Processing Systems*, *33*, 7671-7684.
> > 5. Antoniou, A., Edwards, H., & Storkey, A. (2018, September). How to train your MAML. In *International conference on learning representations*.
> > 6. Alver, S., & Precup, D. (2020). A brief look at generalization in visual meta-reinforcement learning. arXiv preprint arXiv:2006.07262.
> > 7. Melo, L. C. (2022, June). Transformers are meta-reinforcement learners. In *International Conference on Machine Learning* (pp. 15340-15359). PMLR.
> > 8. Shala, G., Biedenkapp, A., & Grabocka, J. (2024). Hierarchical Transformers are Efficient Meta-Reinforcement Learners. *arXiv preprint arXiv:2402.06402*.
> > 9. Grigsby, J., Fan, L., & Zhu, Y. (2023). Amago: Scalable in-context reinforcement learning for adaptive agents. *arXiv preprint arXiv:2310.09971*.

---

### Official Review · Reviewer_788o · 2024-07-23
**Comparison with Craftax needed**

**Rating:** 6
**Confidence:** 4
**Clarity:** The paper is well-written and easy to…

**Review:**

This study is well-motivated and potentially contributes significantly as a benchmark for Meta RL research. The implementation is well thought out, resulting in a concise and user-friendly API. Additionally, the paper is well-organized and very readable. The scalability of the environment produces results consistent with prior research (Jumanji and Pgx), sufficiently demonstrating the implementation's efficiency. Benchmark experiments for Meta-RL show its effectiveness for Meta-RL research purposes.

However, a major drawback is the lack of comparison with recent related research, Craftax (ICML2024). The authors need to discuss comparisons with Craftax.

Another concern is the lack of quantitative comparison between XLand-MiniGrid and the original XLand.
Does XLand-Minigrid actually inherits the key features of XLand?
For example, are methods effective in XLand also useful in XLand-MiniGrid? Can insights gained in XLand-Minigrid be applied to XLand?

---

**I have raised my score from 5 to 6, taking into account the authors' rebuttal.**

**Strengths:**

- The paper is well-motivated, focusing on important issues in Meta-RL research.
- The paper is well-written, readable, and effectively demonstrates its utility.
- The implementation is well-organized.

**Additional Feedback:**

Minor comments include:

- There is a minor formatting issue: The supplementary material should be separated.
- Listing 1 L4: `key = jax.random.key(0)` is not used. It looks strange to me. Authors may review the listing.
- References could be improved. For example, some arxiv papers are already published in proceedings (Jumanji in ICLR2024, Pgx in NeurIPS2023, JaxMARL in AAMAS2024).

Additional feedback that does not affect the score:

Although not mentioned in the paper, the 4-hour execution time for the `trivial-1m` benchmark feels somewhat burdensome to me. As Figure 8 shows, there is a generalization gap even for `trivial-1m`, so a simpler task that completes within an hour could increase visibility.

**Correctness:**

The study provides appropriate experimental support for the usefulness of XLand-MiniGrid.

**Documentation:**

The appendix and repository are well-organized, accurate, and user-friendly.

**Ethics:**

I found no ethics issue.

**Limitations:**

The study lacks sufficient mention of its limitations. While I believe XLand-MiniGrid is useful for Meta RL, there should be some limitations. The authors need to allocate more space to discuss these, either in the conclusion or a separate section.

**Opportunities For Improvement:**

* A discussion on comparisons with Craftax is needed. It is not necessary to be superior in all aspects, but a fair comparison highlighting the advantages, disadvantages, and different use cases is required.
* Also, the paper could be improved by supporting the similarity between XLand-Minigrid and XLand.
* Further discussion on limitations is also needed (see below).

**Improving these concerns could potentially improve my score.**

**Relation To Prior Work:**

The relation to prior work is sufficiently discussed, except for the lack of comparison with Craftax.

**Summary And Contributions:**

This study proposes XLand-MiniGrid, a fast and non-trivial reinforcement learning environment, for Meta-RL research. The environment is entirely implemented in JAX, demonstrating scalability to large batch sizes with high throughput. Baselines for Meta RL are provided, showing the ability to measure train/test generalization performance.

---

> ### Author Rebuttal · Authors · 2024-08-16
>
> We thank the reviewer for detailed feedback and suggested improvements. We provide clarifications below.
>
> > A discussion on comparisons with Craftax is needed. It is not necessary to be superior in all aspects, but a fair comparison highlighting the advantages, disadvantages, and different use cases is required.
>
> Thank you for pointing it out! Craftax is an extremely interesting and related benchmark, and its true that it needs to be mentioned, especially since Craftax itself has a section comparing it to XLand-MiniGrid from its perspective.
>
> We agree with the assessment given in Craftax that XLand-MiniGrid tests broad generalization, while Craftax focuses on deep exploration and achieving a series of sequential goals with increasing complexity. Both benchmarks can be used for open-ended research with limited resources, as Craftax is also fast as it is written in JAX. Craftax provides much more complex and varied world mechanics, but lacks the flexibility to customize the possible challenges or tasks. While in XLand-MiniGrid, thanks to a system of rules and goals, the user can generate any task he or she wants. Therefore, it seems to us that Craftax has the same problems as for example MetaWorld, namely it provides few complex tasks, which is not really suitable for meta-RL research. We try to fill this empty niche and provide a huge distribution of tasks with varying difficulty, from easy to very hard.
>
> We will add the comparison the the Craftax to the already existing related work section (see Appendix D).
>
> > The study lacks sufficient mention of its limitations. While I believe XLand-MiniGrid is useful for Meta RL, there should be some limitations. The authors need to allocate more space to discuss these, either in the conclusion or a separate section.
>
> We are currently discussing limitations in a separate section (see Appendix C) and it seems like we have nothing to cut in the main section to take it out of the appendix. In this section, we go into more detail about the existing limitations regarding MiniGrid, XLand and in general. For example, we state that we don't support multi-agency and procedural generation of worlds as in XLand, or we don't support all types of tiles (lava, moving obstacles) as in MiniGrid. We have also received useful feedback from other reviewers and will improve this section by expanding the discussion. For example, we will add the results of our environment benchmark on low-end GPUs, as the results may be much lower on some of them (T4 GPUs) but quite impressive on others (4090 GPUs) (see [Image](https://ibb.co/P6b1Gxg)).
>
> > Also, the paper could be improved by supporting the similarity between XLand-Minigrid and XLand.
> >
>
> It seems to us that the current version of the paper sufficiently highlights this as it is. We don't hide anywhere what specific parts of XLand we were inspired by, but we do describe the differences we had to make because of the specifics of JAX. Moreover, we try to explain certain decisions, for example, as with hiding rules from observation space by default or referring to the original XLand paper when describing the procedure of task generation.
>
> As for the more specific details and differences, we cover them in the section with limitations (see Appendix C). For example, we do not currently support multi-agent simulations, procedural generation of complex worlds, rules with multiple output entities, or goal composition.
>
> If we're missing something, we'd welcome more specific guidance on which comparisons we should spell out more explicitly.
>
> > There is a minor formatting issue: The supplementary material should be separated.
> >
>
> Thanks, fixed.
>
> > Listing 1 L4: `key = jax.random.key(0)` is not used. It looks strange to me. Authors may review the listing.
> >
>
> Thank you! That was a typo, it should be called `reset_key` and is used to reset the environment.
>
> > References could be improved. For example, some arxiv papers are already published in proceedings (Jumanji in ICLR2024, Pgx in NeurIPS2023, JaxMARL in AAMAS2024).
> >
>
> Thanks! They were updated after the main text was written, so ours are already out of date. We'll fix that.
>
> > Although not mentioned in the paper, the 4-hour execution time for the `trivial-1m` benchmark feels somewhat burdensome to me. As Figure 8 shows, there is a generalization gap even for `trivial-1m`, so a simpler task that completes within an hour could increase visibility.
>
> Indeed. In fact, the complexity of the task is not only determined by the rules and goals, but also by the layout of the map, for example. Maps with 4 rooms and bigger size will be harder due to higher memory and exploration requirements. Thus, we can design a lot simpler task for `trivial-1m` using 1-room 9x9 grid. With such configuration it can be “solved” in under one hour (see [Image](https://ibb.co/6FJrc21)). We also added support for bf16 during training drastically increasing throughput not only on A100/H100 but also on low-end GPUs such as 4090 (see [Image](https://ibb.co/9TH0fqy)). With bf16 enabled it'll take even less, about 30 minutes. We hope that addresses your concern and we will do our best to reflect this in the paper.

---

> > ### Comment · Reviewer_788o · 2024-08-16
> > **A quick reply to the rebuttal**
> >
> > Thank you for the detailed response.
> > As a quick reply, I'll address the points that may take more time to respond to.
> > I will review the other points later.
> >
> > > > Also, the paper could be improved by supporting the similarity between XLand-Minigrid and XLand.
> > > >
> > > It seems to us that the current version of the paper sufficiently highlights this as it is. We don't hide anywhere what specific parts of XLand we were inspired by, but we do describe the differences we had to make because of the specifics of JAX. Moreover, we try to explain certain decisions, for example, as with hiding rules from observation space by default or referring to the original XLand paper
> > > ...
> >
> > What I was expecting here, as I mentioned in the review section (see below), is whether the similarity between XLand and XLand-Minigrid can be **experimentally verified**. I understand that XLand-Minigrid **by design** inherits features of XLand. However, what I want to emphasize is that **inheriting features by design and actually inheriting them in practice are different matters.**
> >
> > > Does XLand-Minigrid actually inherits the key features of XLand? For example, are methods effective in XLand also useful in XLand-MiniGrid? Can insights gained in XLand-Minigrid be applied to XLand?

---

> > > ### Author Rebuttal · Authors · 2024-08-16
> > >
> > > We understand what you mean, but unfortunately we can't test this experimentally and can only speculate, as XLand is not available to regular users and is not open-source.

---

> > > > ### Comment · Reviewer_788o · 2024-08-16
> > > > **A reply to authors**
> > > >
> > > > > We understand what you mean, but unfortunately we can't test this experimentally and can only speculate, as XLand is not available to regular users and is not open-source.
> > > >
> > > > That makes sense. You might want to emphasize the point that XLand is not open-source more prominently in the main text.

---

> > ### Comment · Reviewer_788o · 2024-08-16
> > **Reply to the rebuttal**
> >
> > I have confirmed that all of my other concerns have been addressed.
> > The comparison with Craftax was convincing.

---

> > ### Comment · Reviewer_788o · 2024-08-16
> > **A response to the rebuttal**
> >
> > Overall, the rebuttal was convincing. If the paper is accepted, there should be an additional page of space available, so I recommend moving the comparison with Craftax and some of the important discussions from the appendix to the main text.

---

> > > ### Author Rebuttal · Authors · 2024-08-16
> > >
> > > Thank you for the provided feedback! We will do our best to address all of the above in the final version.

---

### Official Review · Reviewer_R62J · 2024-07-23
**High-speed, Jax environment for meta reinforcement learning**

**Rating:** 7
**Confidence:** 4
**Clarity:** Paper is well written and well struct…

**Review:**

The presented environment enables smaller research groups with less access to compute to experiment with large-scale-like tasks, similar to Deepmind XLand/Ada work, as well as generalization, similar to ProcGen. Groups with more compute can also use the environment to run large-scale experiments (e.g., large hyperparameter sweeps, very long trainings). I consider these a significant-enough contribution to the community, as RL agents still lack in generalization aspect.

The environment lacks many features that could make it more complete (as noted by the authors), and the current work only contains one agent baseline results. There is also a chance the task is too specific to contribute meaningful signal to method development, considering the more complex version was solved by Deepmind Ada. However, this environment will still be useful in the set of environments that allow more fundamental study of the algorithms and their limitations.

* Quality: High. Implementations are of high quality, and paper has benchmarks to show the speed + baseline results.

* Clarity: Clear paper.

* Originality: Medium. By design, XLand-MiniGrid is a combination of existing things, but XLand was not open sourced. However, bringing them together in a simple environment is somewhat novel.

* Significance: Medium. Environment for mainly benchmarking meta-RL algorithms at a fast rate (given the amount of training meta-RL requires), but only does so in a simplified environment. However, this allows initial experimentation and more fundamental study of the methods, and the environment also provides more standardized MiniGrid tasks in faster format.

**Strengths:**

* Fast environment for Meta-RL, with design choices that allow it to be used for these purposes over some existing solutions.
* Brings the tasks defined in a landmark paper (Deepmind Ada) in a smaller scale for the community to experiment with.
* Defines benchmarks with different difficulties, and provides baseline results with the benchmarks.
* Baseline results show that the challenge, despite living in a simple grid-world, is still challenging for the baseline solution.

**Additional Feedback:**

-

**Correctness:**

- Experiments were ran over at least three seeds. While more would be desirable, the variance is small enough to make results significant.
- Includes train/test experiments (Figure 8), in addition to more traditional RL setup of testing on the train set.
- Long training run to confirm the solution is simply not just matter of running longer.

**Documentation:**

- Code is available on github
- Examples provided as jupyter notebooks
- Code for running the experiments is provided in the github repository

**Ethics:**

No ethical concerns.

**Limitations:**

* As noted by authors, in the current form the XLand-MiniGrid is lacking features that either XLand or MiniGrid has, but granted, XLand is not released. Similarly, grid-world nature of the environment restricts the complexity, but this is by design to keep the environment fast.
* Also noted by the authors, using Jax to implement the environment limits the flexibility of what can be effeciently implemented

**Opportunities For Improvement:**

* One major weakness is only having one algorithm for baseline results. While this is a benchmark paper, and a single solution is already an indicator of the difficulty, it is especially weird seeing in comparison to Ada and how it "solved" much of the more difficult XLand. This is likely due to Ada being more complex than RL^2, but considering Ada solved a more complex setting, it makes me doubt the value of this benchmark (i.e., there may already exist a solution to this problem).

### Questions / Requests

1) Can you add benchmarks with more commodity hardware? While high speeds on an A100 are impressive, a fast and light library will also allow people with access to weaker hardware to run good experiments like done in this paper. Unfortunately access to A100 (or similar GPU) is a rather high bar.
2) What are the upper bounds for agent performance in each of the benchmarks? It is hard to say if the agent is nearing any optimal performance with return of "20" or "10". I recommend normalizing plots between minimum and maximum achievable performance, or at least reporting the optimal performance number somwhere close to the plots.
3) As pointed out by the authors, RL^2 does not reach the "optimal" results. Would you expect the results change with a more modern meta-learning method, or with a larger network? Somehow I am bit skeptical that with 1T transitions the algorithm did not learn, but was trending to improve. Perhaps some hyperparameter-tuning would have solved the issue? Is there a chance to include some other meta-RL method?

**Relation To Prior Work:**

- Authors compare the solution to the inspiration (XLand and MiniGrid), as well the attempts to use Jax to speed up environments or moving environments completely on GPU.
- Compares to existing Jax libraries and highlights shortcomings in them that make them inpractical for Meta-RL.
- Authors also compare to previous meta-learning environments. The proposed environment is especially fitting due to its fast speed, as Meta-RL requires plenty of data. Specifically, they bring good sides of Alchemy environments without the speed sacrifice (Alchemy was implemented in Unity and ran at 30 FPS)

**Summary And Contributions:**

Authors present XLand-MiniGrid environment for fast experimentation with meta reinforcement learning (RL). XLand-MiniGrid implements the task and goal generation of XLand environment, which has not been released, with MiniGrid-like grid environment, allowing users to run environment at millions of steps per second on a GPU. The XLand-like benchmarks generate hierarchical set of goals agent has to complete, where each step usually produces certain object, and agent has to follow this follow tree of producing all items it needs to finish the task.

---

> ### Author Rebuttal · Authors · 2024-08-16
>
> We thank the reviewer for the time spent on the review, as well as for the positive feedback. We tried to address the concerns below.
>
> > One major weakness is only having one algorithm for baseline results. While this is a benchmark paper, and a single solution is already an indicator of the difficulty, it is especially weird seeing in comparison to Ada and how it "solved" much of the more difficult XLand. This is likely due to Ada being more complex than RL^2, but considering Ada solved a more complex setting, it makes me doubt the value of this benchmark (i.e., there may already exist a solution to this problem).
>
> These are all valid concerns. However, our goal was not and is not to create a new grand benchmark (like NetHack) that would be completely out of reach for the most advanced algorithms with the almost infinite resources available (as is the case with Ada from DeepMind). Benchmarks we provide are done more out of convenience and it is likely that they can be solved with comparable resources to Ada. However, most academics do not have such resources and therefore the current benchmarks are complex enough to be interesting and instructive. Our main aim was to provide academics with a platform to effectively test ideas and be creative on a scale not previously available to them and we believe we have successfully accomplished this goal. After all, not all research is about achieving a new state-of-the-art. Moreover, a benchmark does not lose its value if it is solved once (e.g. Atari).
>
> > Can you add benchmarks with more commodity hardware?
>
> Indeed, given that our main focus is on democratizing the meta-RL field, we should have given this more attention. We've tried our best to complement the benchmarks with more affordable GPUs like the 4090, 3090 and "free-for-all" T4s from Google Colab or Kaggle. We will add these figures to the appendix as a separate section. See [Image 1](https://ibb.co/LRts4yq) and [Image 2](https://ibb.co/P6b1Gxg).
>
> As can be seen, pure throughput with random policy is decent on all GPUs, even the oldest T4 is capable of achieving millions of steps per second and scales with more environments. However, during PPO training the difference becomes more apparent. The maximum SPS for a T4x2 is around 160k and for 3090 is around 400k and for 4090 is around 800k. To compare, A100 is able to achieve 1.0M and H100 even higher 1.2M SPS. The most important factor here is how fast we can get through a single epoch of training on the batch collected from all environments,which in turn comes down to the maximum mini-batch size. Ideally, for every increase in the number of parallel environments, we should also increase the mini-batch size. If this does not happen, saturation occurs. Due to the smaller memory capacity of consumer GPUs the saturation during training comes a lot earlier than for A100/H100. However, 800k on 4090 is still a good result, more than any other benchmark (like MetaWorld, Alchemy, etc) can achieve with such resources.
>
> To make training even more accessible on the next generation of GPUs, we added support for bf16 during training. On our hardware and with same hyperparamers it increased throughput dramatically on H100 and 4090 GPUs, reaching 1.5M and 2.1M respectively (see [Image 3](https://ibb.co/9TH0fqy)).
>
> > What are the upper bounds for agent performance in each of the benchmarks?... I recommend normalizing plots between minimum and maximum achievable performance, or at least reporting the optimal performance number somewhere close to the plots.
>
> Actually, in the first version of the paper, we normalized all the results due to this exact reason. However, while this is an extremely tempting idea, it has a few drawbacks.  First of all, given the diversity and variation in task complexity, we do not have any tool to reliably know the optimal policy. One could use human performance, but it is impossible to label up millions of tasks by hand. What other proxies we have?
>
> We used goal-conditioned policies. Theoretically, they can zero-shot solve new tasks given their ground-truth descriptions. However, we quickly realized that this normalization was misleading as to the underlying performance. For example, in a complex task, a meta-RL  agent would have to spend some of the given 25 tries just to figure out what to do. There is no way to do this without it. However, a goal-conditioned agent will be able to solve the task zero-shot in the first episode, and get a total of 25 rewards (for each subsequent episode). Thus, even an optimal meta-RL agent will not be able to achieve 1.0 normalized score. So we opted not to normalize. We'd welcome other suggestions or ideas, it's quite possible we've missed something.
>
> > Would you expect the results change with a more modern meta-learning method, or with a larger network? Perhaps some hyperparameter-tuning would have solved the issue?
>
> We actually tuned the parameters quite a bit (considering how much cheaper it is to do it at this speed) and didn't see much of a gain, especially when looking at generalization or lower bound performance like 20-percentile. We have not yet release-ready evidence that when replacing the GRU with the Transformer-XL, the results come out much better on average, but the near-zero 20-percentile problem remains.
>
> > Is there a chance to include some other meta-RL method?
>
> Given that there are no libraries for meta-RL with tested algorithms like CleanRL, stable-baslines3, torchrl and others, and implementation of algorithms in meta-RL is much more complicated (especially so that it is compatible with jit in JAX), we decided that it is much more important to provide a single but high-quality and fast implementation of an algorithm for which there are evidences that it can solve very complex problems if sufficiently scaled up. This gives a much more convenient starting point for practitioners. We hope to add more algorithms in the future.

---

> > ### Comment · Reviewer_R62J · 2024-08-23
> >
> > I thank the authors for very detailed rebuttal! I am inclined to keep my rating as is (7); while the rebuttal improved my view of the paper, the lack of more baselines or experiments showing how this is important for the prevents me from increasing the rating with good conscience. To showcase importance, there could be a experiment/result that shows how this benchmark adds challenges or dimensions of challenge that have not seen before (arguably, the XLand-like setup is like this, but without experiment it is hard to argue with).
> >
> > Some detailed replies below.
> >
> > > One baseline / Another Meta-RL implementation
> >
> > While you could also implement the methods yourself (which is dangerous in on itself), I agree that it was better to use the time to delve deeper into the environment performance and how results change when increasing the environment difficulty.
> >
> > > Moreover, a benchmark does not lose its value if it is solved once (e.g. Atari).
> >
> > This is a good point I did not consider. A faster environment may even be better, as it allows to properly validate the ideas and hypothesis, instead of focusing on one narrow path of solutions that leads to "landmark" result. Thank you for bringing this up (very refreshing among all the state-of-the-art papers).
> >
> > > GPU results (4090, 3090, T4)
> >
> > Thank you for sharing the pictures (taking the extra mile to provide the images instead of just promising to add them). These results are interesting already in compute wise: gaming GPUs (3090, 4090) are faster in raw throughput, probably thanks to higher clocks, but with training A100 and  H100 still dominate. In any case, this indeed shows that you can train agents fast on more commodity hardware.
> >
> > > Normalization of the results
> >
> > Thank you for the detailed explanation. At the time of the writing I might have believed there is some heuristic way to compute the minimum number of steps required to solve a task (for example), given it is a gridworld and a combination of tasks. However on a deeper thought, the composition of rules makes things tricky. With this, I understand not having normalization. I would still welcome having some sense of lower or upper boundaries, or at least including this explanation in the paper for future readers.
> >
> > > Larger network and hyperparameter tuning
> >
> > Thank you for the answer. You may want to consider highlighting this in the main paper around the results, so that readers do not pose the same question.

---

### Official Review · Reviewer_Q9K1 · 2024-08-06
**Comments for paper #1574**

**Rating:** 8
**Confidence:** 4
**Clarity:** Very well written paper.

**Review:**

pros:
1. Very well-written and easy-to-follow paper. The clarification of each parts of the proposed XLand-MiniGrid, including the motivation and high-level design logic, are properly orgnized.
2. Though inspried from previous works and showing certain similarity, XLand-MiniGrid economically combines useful designs of these previous works and is still itself a novel benchmark suite. In particular, the proposed Rules and Goals which can be easily integrated into grid  environments provide a novel aspect of producing good training samples for Meta-RL. Besides, development efforts spent on Jax coding ensure the overall efficiency.
3. The target domain of this paper is Meta-RL, which holds a wide range of audience. This makes this work a significant contribution.

cons:
1. Missing throughoutput validation on low performance GPU, e.g., 4090, 3090.
2. Missing the comparison with the previous works. In particular, the results in Figure 8 can only demonstrate that XLand-MiniGrid is challenging. I wonder the results in Figure 8 for  MiniGrid. The performance drop ratio between the train and test would be a good metric to measure how good is XLand-MiniGrid compared with MiniGrid.

------
All concerns have been addressed.

**Strengths:**

see pros in Review.

**Additional Feedback:**

N/A

**Correctness:**

Yes, I believe the claims of this paper are correctly addressed. The construction of the environments are pre-processed to avoid large computational budget for users, which is rational. The evaluation provided ar Experimrnts follows regular RL settings, which is correct and convincing.

**Documentation:**

Yes.

**Limitations:**

I would appreciate the authors if they could provide a Limitation discussion in their paper. This would makes the paper better. e.g., 1) whether the proposed toolkit underperforms for some realistic Meta-RL scenarios since the grid environments are still simpler than these scenarios. 2) For those researchers who can not access high performance GPU such A100, is XLand-MiniGrid still capable providing the same training quality and efficiency?

**Opportunities For Improvement:**

see cons in Review.

**Relation To Prior Work:**

Yes, the relationship, including the similarity and difference are fully addressed in Appendix.

**Summary And Contributions:**

This paper introduces XLand-MiniGrid, a toolkit providing dynamic grid-environments for Meta-RL.

[Primary Contributions]
1. Novel Meta-RL environment toolkit, with a large number of diverse environments.
2. Well-equipped with JAX, supporting high-efficiency rollout.
3. Solid validation for the generalization difficulty of the generated environments, which might potentially promote the development of Meta-RL method.

---

> ### Author Rebuttal · Authors · 2024-08-16
>
> We thank the reviewer for the time spent on the review and the suggested improvements. We tried to address them below.
>
> > Missing throughoutput validation on low performance GPU, e.g., 4090, 3090.
>
> Indeed, given that our main focus is on democratizing the meta-RL field, we should have given this more attention. We've tried our best to complement the benchmarks with more affordable GPUs like the 4090, 3090 and "free-for-all" T4s from Google Colab or Kaggle. We will add these figures to the appendix as a separate section. See [Image 1](https://ibb.co/LRts4yq) and [Image 2](https://ibb.co/P6b1Gxg).
>
> As can be seen, pure throughput with random policy is decent on all GPUs, even the oldest T4 is capable of achieving millions of steps per second and scales with more environments. However, during PPO training the difference becomes more apparent. The maximum SPS for a T4x2 is around 160k and for 3090 is around 400k and for 4090 is around 800k. To compare, A100 is able to achieve 1.0M and H100 even higher 1.2M SPS.  The most important factor here is how fast we can get through a single epoch of training on the batch collected from all environments, which in turn comes down to the maximum mini-batch size. Ideally, for every increase in the number of parallel environments, we should also increase the mini-batch size. If this does not happen, saturation occurs. Due to the smaller memory capacity of consumer GPUs the saturation during training comes a lot earlier than for A100/H100. However, 800k on 4090 is still a good result, more than any other benchmark (like MetaWorld, Alchemy, etc) can achieve with such resources.
>
> To make training even more accessible on the next generation of GPUs, we added support for bf16 during training. On our hardware and with same hyperparamers it increased throughput dramatically on H100 and 4090 GPUs, reaching 1.5M and 2.1M respectively (see [Image 3](https://ibb.co/9TH0fqy)).
>
> > Missing the comparison with the previous works. In particular, the results in Figure 8 can only demonstrate that XLand-MiniGrid is challenging. I wonder the results in Figure 8 for MiniGrid. The performance drop ratio between the train and test would be a good metric to measure how good is XLand-MiniGrid compared with MiniGrid.
>
> We're sorry if it wasn't clear from the paper, but MiniGrid and XLand-MiniGrid have a completely different focus that can't be compared directly. MiniGrid is a set of environments for regular single-task RL and the results on it are known from many already existing papers. We didn't have the goal of making MiniGrid more complex, on the contrary, we tried to port it exactly.
>
> We implemented the MiniGrid-benchmark environments only to show the versatility and flexibility of our “core” gridworld system. Thus, even practitioners not interested in meta-RL can nevertheless benefit from a significant acceleration of their experiments on MiniGrid using our environments as they are still based on JAX and leverage GPU power.
>
> > I would appreciate the authors if they could provide a Limitation discussion in their paper. e.g., 1) whether the proposed toolkit underperforms for some realistic Meta-RL scenarios since the grid environments are still simpler than these scenarios. 2) For those researchers who can not access high performance GPU such A100, is XLand-MiniGrid still capable providing the same training quality and efficiency?
>
> Thanks for the suggestion! We will expand our already existing Limitations section (see Appendix C), adding more discussion around the relation of proposed benchmark to real world and training efficiency on low-end GPUs.

---

> > ### Comment · Reviewer_Q9K1 · 2024-08-16
> >
> > Thanks for the response. In particular, I am satisfied with the explanation of why not compare the previous works. I suggest the authors add the refinement into their updated paper. Since all concerns have been addressed, I will rise my score to 8.

---

### Decision · Program_Chairs · 2024-09-26

**Decision:**

Accept (Poster)

**Comment:**

This paper introduces XLand-MiniGrid, a suite of grid-world environments designed for meta-reinforcement learning (Meta-RL) research, implemented using JAX for high throughput and scalability. The environments combine the simplicity of MiniGrid with the diversity of XLand tasks, allowing researchers to test Meta-RL algorithms efficiently on large-scale benchmarks. The paper emphasizes the platform's potential for democratizing access to large-scale experiments with limited resources.

Strengths:
- The use of JAX enables fast execution, supporting millions of steps per second even on consumer-grade GPUs, which is a significant contribution for researchers with limited access to high-end hardware.
- The suite provides diverse tasks with varying difficulty, enabling thorough evaluation of Meta-RL algorithms.
- The authors provide well-documented open-source code, allowing the community to easily access and use the platform.
- The paper is well-written and easy to follow, with a clear explanation of the design and implementation choices.

Weaknesses:
- Reviewers noted that only one baseline (RL²) is tested, which limits the paper’s capacity to validate the benchmark’s general suitability for Meta-RL. A more comprehensive evaluation with additional Meta-RL methods would have strengthened the paper.
- While comparisons with other environments are mentioned, they are mainly in the appendix. Reviewers suggested including more direct experimental comparisons with existing benchmarks like Craftax or MiniGrid in the main text to better highlight the advantages of XLand-MiniGrid.
- One reviewer pointed out that while the benchmark provides diverse tasks, it lacks the depth and complexity of environments like XLand. This could limit its applicability for some advanced Meta-RL scenarios.
- Some reviewers felt that the paper could benefit from a more detailed discussion of its limitations, particularly concerning task complexity and hardware requirements for optimal performance.

Overall assessment: Despite some limitations in the scope of experimentation, the paper provides a valuable contribution to the Meta-RL research community by offering a scalable, high-throughput benchmark platform. The open-source nature and user-friendly design will make it a useful tool for a wide range of researchers.